# Highly efficient and salt rejecting solar evaporation via a wick-free confined water layer

Lenan Zhang[1,3], Xiangyu Li[1,3], Yang Zhong [1], Arny Leroy [1], Zhenyuan Xu[2], Lin Zhao [1] &
Evelyn N. Wang [1✉]

Recent advances in thermally localized solar evaporation hold significant promise for vapor generation, seawater desalination, wastewater treatment, and medical sterilization. However, salt accumulation is one of the key bottlenecks for reliable adoption. Here, we demonstrate highly efficient (>80% solar-to-vapor conversion efficiency) and salt rejecting (20 weight % salinity) solar evaporation by engineering the fluidic flow in a wick-free confined water layer. With mechanistic modeling and experimental characterization of salt transport, we show that natural convection can be triggered in the confined water. More notably, there exists a regime enabling simultaneous thermal localization and salt rejection, i.e., natural convection significantly accelerates salt rejection while inducing negligible additional heat loss. Furthermore, we show the broad applicability by integrating this confined water layer with a recently developed contactless solar evaporator and report an improved efficiency. This work elucidates the fundamentals of salt transport and offers a low-cost strategy for high-performance solar evaporation.

[1] Department of Mechanical Engineering, Massachusetts Institute of Technology, Cambridge, MA 02139, USA. [2] Institute of Refrigeration and Cryogenics Shanghai Jiao Tong University, Shanghai 200240, China. [3]These authors contributed equally: Lenan Zhang, Xiangyu Li. ✉email: enwang@mit.edu

Water scarcity has become a severe challenge for humanity since two-thirds of the global population is affected by water shortage[1]. Owing to the significant potential for clean water production, highly efficient solar evaporation by localizing the solar-thermal conversion process near the evaporating interface has attracted tremendous research interest in passive vapor generation[2–7], seawater desalination[8–13], wastewater treatment[14–18], and medical sterilization[19]. However, due to the ultralow diffusivity of salt in water (~$10^{-9}$ m$^2$ s$^{-1}$, as a reference in comparison to the diffusivity of vapor in air is ~$10^{-5}$ m$^2$ s$^{-1}$), there is significant salt accumulation, which induces undesirable fouling, reduces evaporation rate, and degrades device reliability. This effect has become one of the key practical challenges for a range of applications[8,20–26].

To enable highly efficient and reliable solar evaporation, a key bottleneck is to achieve simultaneous thermal localization and salt rejection. In a typical thermally localized solar evaporation device (Fig. 1a), a capillary wick structure is used to enable the solar-thermal conversion, thermal localization, and passive water supply[3,5,7,8,20]. However, the use of a wick structure creates an extra transport resistance for salt diffusion such that clogging or crystallization unavoidably occurs on the highly confined water

interface under the intense solar flux or after long-term operation (Fig. 1a)[20,26]. Although several recent strategies such as separation by function[21,27–30] and enhanced diffusion by macroscopic pores[8,18,31] have been applied to wick structure-based devices, the salt accumulation, especially when evaporating high-salinity brine, still cannot be fully addressed. On the other hand, contactless solar evaporation, converting sunlight into infrared (IR) thermal radiation with a non-contact solar absorber, directly localizes heat on the bulk water interface and therefore completely eliminates the use of a wick structure (Fig. 1b)[16,32]. Superior salt rejection has been recently demonstrated with high-salinity brine (up to 25 weight % (wt%)). Furthermore, since the vapor temperature is not pinned by the liquid-vapor interface, contactless solar evaporation is capable of producing superheated steam, which is promising for high-temperature applications such as solar sterilization[32]. However, due to the increased heat loss through the non-confined bulk water (Fig. 1b), the highest reported solar-vapor conversion efficiency of contactless solar evaporation (43%) is lower than that of conventional wick structure-based devices (>60%)[3,16,20].

Therefore, the tradeoff between thermal localization and salt rejection (Fig. 1a, b) highlights the opportunity space with a

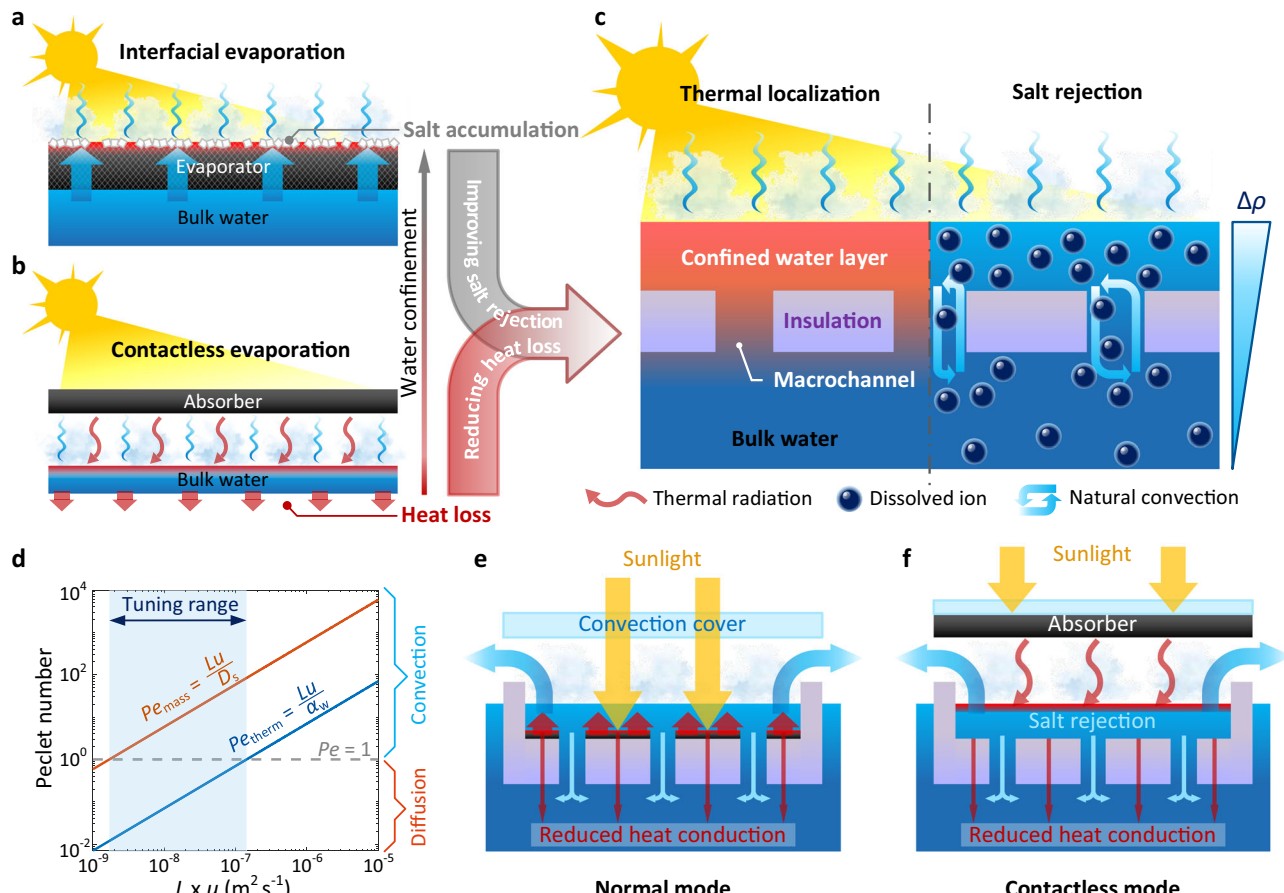

**Fig. 1 Wick-free confined water layer for solar evaporation. a** Interfacial evaporation enabled by wick structure-based solar evaporator. Salt accumulation commonly occurs on the water–air interface due to the strong water confinement. **b** Contactless evaporation by separating the solar absorber and bulk water with an air gap. Heat loss through the bulk water increases due to the absence of water confinement. **c** Simultaneous thermal localization and salt rejection via a wick-free confined water layer structure. Heat loss is impeded by a self-floating thermal insulation layer. Macrochannels are created to initiate the natural convection. Salt rejection is driven by natural convection due to the density gradient across the confined water layer and bulk water. **d** Thermal ($Pe_{therm}$) and mass ($Pe_{mass}$) transport Peclet numbers as a function of the flow condition ($L \times u$). Convection-dominated process occurs when the Peclet number is larger than one, otherwise the transport process is governed by diffusion. The blue band indicates the tuning range of fluidic flow to realize the simultaneous thermal localization and salt rejection. **e, f** Applications of the confined water layer structure to the normal mode (**e**) and contactless mode (**f**) evaporators.

moderate amount of water confinement (Fig. 1c). Instead of pursuing extreme interfacial water confinement using wick structures or eliminating water confinement with contactless heating, a few very recent bio-inspired designs, confining a thin water layer (~mm thickness) within a 3D printed conic structure[24] or hydrophobic porous absorber[17], demonstrated significant enhancement in salt rejection. Despite these efforts, a pathway toward simultaneous thermal localization and salt rejection in a simple evaporator has remained elusive because the mechanics of salt transport during solar evaporation are not well-understood[8,20,26]. Specifically, there are two fundamental problems to be addressed: (1) Breaking the conventional diffusion-limited salt rejection by engineering passive convective flow. Since significant enhancement of salt rejection can be achieved by introducing convective flow (e.g., Marangoni flow[22] and unidirectional flow[23]), a quantitative understanding of how to passively initiate a convective flow becomes important. (2) Understanding the interplay between thermal localization and salt rejection due to water confinement and convective flow. Since convective flow also increases heat loss, a guideline to maximize salt rejection while minimizing heat loss is required[20]. In addition, as a practical consideration, it is also essential to develop a simple solar evaporator with fewer material restrictions and lower cost[8,20].

Here, we show a highly efficient and salt rejecting solar evaporation approach by engineering the convective flow in a confined water layer. With a mechanistic model coupling the salt transport with the fluidic flow and heat transport, we show that gravity-driven natural convection can be passively triggered by carefully engineering the geometry of the confined water layer. More importantly, taking advantage of the two orders of magnitude difference in the mass diffusivity of salt in water and thermal diffusivity of water, we demonstrate a regime where natural convection dominates the salt rejection while having a negligible impact on heat loss. Simultaneous thermal localization and salt rejection were experimentally demonstrated, which agrees well with our theoretical prediction. We show the improved performance in both laboratory and outdoor conditions. Owing to the superior thermal localization, above 90% solar-to-vapor conversion efficiency was demonstrated, which is comparable to the performance of state-of-the-art wick structure-based evaporators. Meanwhile, evaporating high-salinity brine up to 20 wt% without salt crystallization was achieved. Stable evaporation rate and salt rejection were also confirmed through a 1-week reliability test. Furthermore, to show the broad applicability of the proposed design, we integrated the confined water layer with a contactless solar evaporator, which is important for high-temperature applications[32] and wastewater treatment[16]. We report an improved solar-to-vapor conversion efficiency (≈50%) for contactless solar evaporation. This work demonstrates high-performance solar evaporation by leveraging the confined water layer and convective flow and eliminates the necessity of wick structures, consequently relaxing materials restrictions. The comprehensive modeling and characterization elucidate the underlying physics of salt transport during solar evaporation, which can serve as general design guidelines for highly efficient and salt rejecting evaporators.

## Results

**Wick-free self-floating confined water layer**. Figure 1c shows the concept of simultaneous thermal localization and salt rejection via a confined water layer. The water confinement is realized by a neutrally buoyant thermal insulation. The top water layer is connected with bottom bulk water by vertical macrochannels

through the thermal insulation (Fig. 1c). Due to the neutral buoyancy and macrochannel connection, the self-floating thermal insulation moves synchronously with the water–air interface, maintaining a stable confined water layer throughout the entire evaporation process without the need for wicking. Since solar-thermal conversion is localized within the water layer and heat loss to the bulk water is blocked by the thermal insulation, high thermal localization, comparable to the wick structure-based solar evaporation, can be achieved (left panel, Fig. 1c). Meanwhile, evaporation leads to higher salt concentration close to the water–air interface, which creates a density gradient across the confined water layer (right panel, Fig. 1c). Therefore, with proper design of the macrochannel size, it is possible to initiate natural convection along the density gradient and accelerate salt transport from the top water layer to the bottom bulk water.

Figure 1d shows the opportunity space of natural convection by taking advantage of the drastic difference in heat and mass transport characteristics. The contribution of convection relative to diffusion is described by the Peclet number, which is defined by the ratio of the product of characteristic length $L$ and characteristic flow velocity $u$ to the thermal or mass diffusivity. Due to the two orders of magnitude difference in the mass diffusivity of salt in water ($D_s \sim 10^{-9}$ m$^2$ s$^{-1}$) and thermal diffusivity of water ($\alpha_w \sim 10^{-7}$ m$^2$ s$^{-1}$), the mass transport Peclet number ($Pe_{mass} = L \times u/D_s$) is much larger than the thermal transport Peclet number ($Pe_{therm} = L \times u/\alpha_w$) under the same flow condition ($L \times u$). Therefore, there is an optimal tuning range of the fluidic flow (blue band in Fig. 1d) where the salt transport through the macrochannels is driven by natural convection ($Pe_{mass} > 1$) while the heat loss from the confined water layer to the bulk water is still governed by conduction ($Pe_{therm} < 1$), indicating the possibility of simultaneous thermal localization and salt rejection. We first integrated the confined water layer structure with a normal mode evaporator, where the solar absorber is located at the interface of the top water layer and thermal insulation (Fig. 1e). We tested the normal mode evaporator because it is widely used for desalination, while requiring improved salt rejecting performance. We also applied it to the contactless mode evaporator, where the solar absorber is above the confined water layer and IR absorption occurs at the water–air interface (Fig. 1f). We tested the contactless mode evaporator because it is promising for the high-temperature steam generation and wastewater treatment but requires improved solar-to-vapor conversion efficiency.

Figure 2a and b shows a prototype of the wick-free self-floating confined water layer structure, which consists of an insulating ring (2.5 mm thickness and 6 mm height), a solar absorber (31 mm diameter), a floating insulation (36 mm diameter), five macrochannels, and a balancing weight (see "Methods", Supplementary Note 1, and Supplementary Figs. 1 and 2 for details of device design and fabrication). We used low-cost and commercially available materials for this prototype. Low thermal conductivity polystyrene and polyurethane foams (<0.03 W m$^{-1}$ K$^{-1}$ thermal conductivity) were used as the insulating ring and floating insulation, respectively. For the normal mode evaporator, black paint was sprayed on the top of the floating insulation to achieve 95.3% solar absorptance (see "Methods" and Supplementary Fig. 3 for details of the solar absorptance characterization). For the contactless mode evaporator, the solar absorber above the floating structure was painted black instead. Note that we used copper plates as the balancing weight in this prototype owing to its high density and ease of fabrication. However, for large-scale applications, many low-cost and high-density materials, such as concrete and brick, can be used. We estimate that the total material cost of a wick-free self-floating confined water layer structure is ≈$2.5–3.9 m$^{-2}$

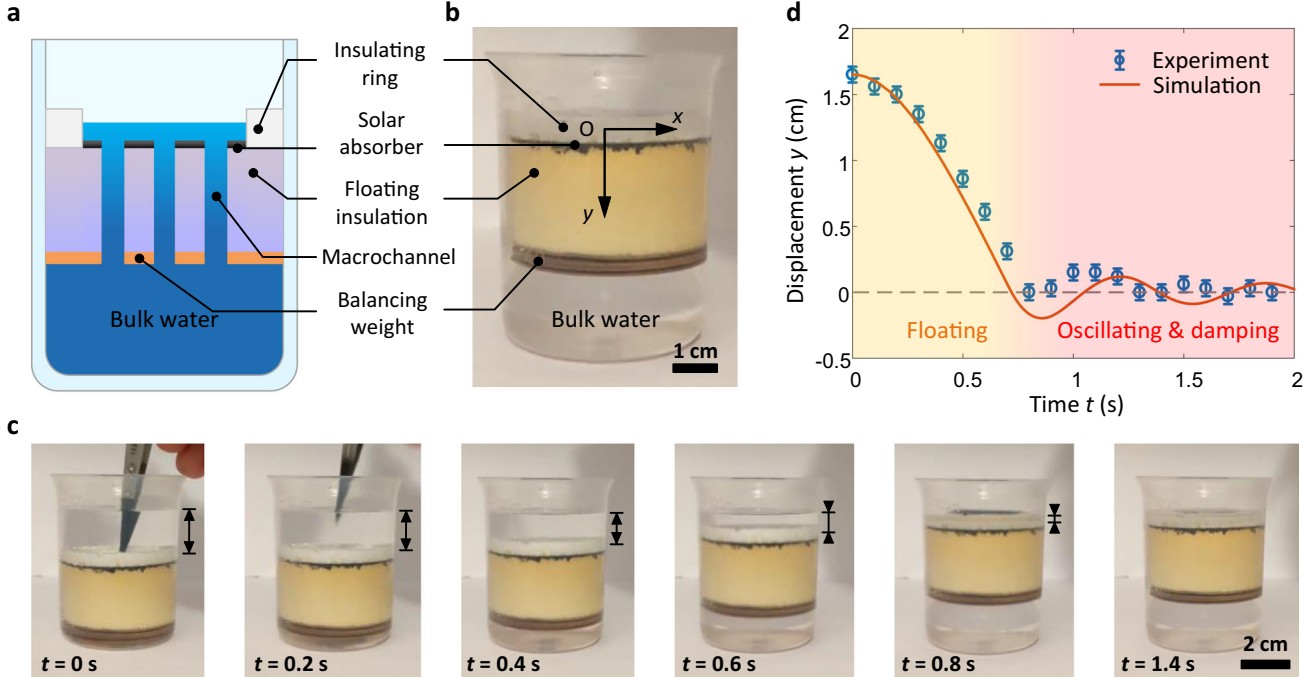

**Fig. 2 Prototype of the confined water layer structure.** Schematic (**a**) and image (**b**) of the fabricated prototype comprising of an insulating ring, a solar absorber, a floating insulation, five macrochannels, and a balancing weight. **c** Time-lapse images showing the dynamic response of the confined water layer structure under extreme displacement. **d** Confined water layer displacement y as a function of time t extracted from time-lapse images (blue data points) and simulated by a buoyancy-driven damped oscillator model (red curve). The thermal insulation layer experienced a rapid floating process (yellow regime) followed by a weak oscillating and damping process (light red regime). Error bars were determined by the spatial and temporal resolution of time-lapse images.

using concrete as the balancing weight (see Supplementary Note 2 for cost analysis). With a careful design of the balancing weight, a stable water layer (≈5 mm thickness) was maintained on the floating insulation (Fig. 2a, b), enabling thermal insulation, salt rejection, and passive water supply.

We tested the mechanical stability of the floating structure, which is critical for realistic operating conditions. We applied an extreme displacement (≈1.6 cm out of the equilibrium position) on the floating structure by pushing it to the bottom of the reservoir and recorded its dynamic response (Fig. 2c). Figure 2d shows the fast dynamic response of the floating structure. The floating structure rapidly moved back to the water–air interface within ≈1 s (yellow regime in Fig. 2d) and then slightly oscillated around the equilibrium position with damping (light red regime in Fig. 2d). This dynamic response can be well-described by a buoyancy-driven damped oscillator (red curve in Fig. 2d, see "Methods" for details of the dynamics response simulation).

**Modeling of salt transport in the confined water layer.** To understand the fundamentals of salt transport, we developed a mechanistic model by coupling it with fluidic flow and heat transport. The flow field is described by mass conservation (Eq. 1) and momentum conservation, Navier-Stokes (Eq. 2) equations, for time-dependent incompressible flow,

$$\nabla \cdot \mathbf{u} = 0 \tag{1}$$

$$\rho \frac{\partial \mathbf{u}}{\partial t} + \rho(\mathbf{u} \cdot \nabla)\mathbf{u} = -\nabla p + \nabla \cdot [\mu(\nabla \mathbf{u} + (\nabla \mathbf{u})^{\mathrm{T}})] + \rho \mathbf{g} \tag{2}$$

where $\mathbf{u}$, $p$, and $\mathbf{g}$ are the vector flow field, pressure, and gravitational acceleration, respectively. To capture the natural convection effect, the salt concentration $c$ and temperature $T$ dependent brine density $\rho = \rho(c, T)$ was applied to Eq. (2) (see Supplementary

Note 3 for details). The heat and salt transport are described by the convection-diffusion equations (Eqs. 3 and 4),

$$\frac{\partial T}{\partial t} - \nabla \cdot (\alpha_{\mathrm{w}} \nabla T) + \mathbf{u} \cdot \nabla T = 0 \tag{3}$$

$$\frac{\partial c}{\partial t} - \nabla \cdot (D_{\mathrm{s}} \nabla c) + \mathbf{u} \cdot \nabla c = 0. \tag{4}$$

Note that Eqs. (1–4) are fully coupled and need to be solved simultaneously because the fluidic flow $\mathbf{u}$ is driven by $\rho(c, T)$ while the distribution of $c$ and $T$ is determined by $\mathbf{u}$. A uniform heat flux due to the solar-thermal conversion was applied to the bottom of the confined water layer, which was determined by the incident solar flux and absorptance of the solar absorber. Meanwhile, an evaporative flux was applied to the water–air interface. The evaporative flux was prescribed by a mass transfer coefficient, which was determined from the experimental calibration. The accumulated salt flux at the water–air interface can thus be converted from the evaporative flux and brine salinity, which was used as the boundary condition for Eq. (4) (see "Methods", Supplementary Note 3, and Supplementary Figs. 4 and 5 for details of the salt transport model).

**Natural convection enhanced salt rejection.** Figure 3a shows the model predicted heat and mass transport characteristics of the confined water layer as a function of the macrochannel diameter $d$. We used the salinity and temperature difference between the inlet (on the confined water layer side) and outlet (on the bulk water side) of the macrochannel to quantify the salt rejection and thermal localization capability, respectively. For the heat transport process, the temperature difference $\Delta T$ initially slowly decreased with $d$. This slight reduction in thermal localization is mainly attributed to the decrease of heat conduction resistance (light red regime in Fig. 3a). When $d$ increased to ≈3.5 mm,

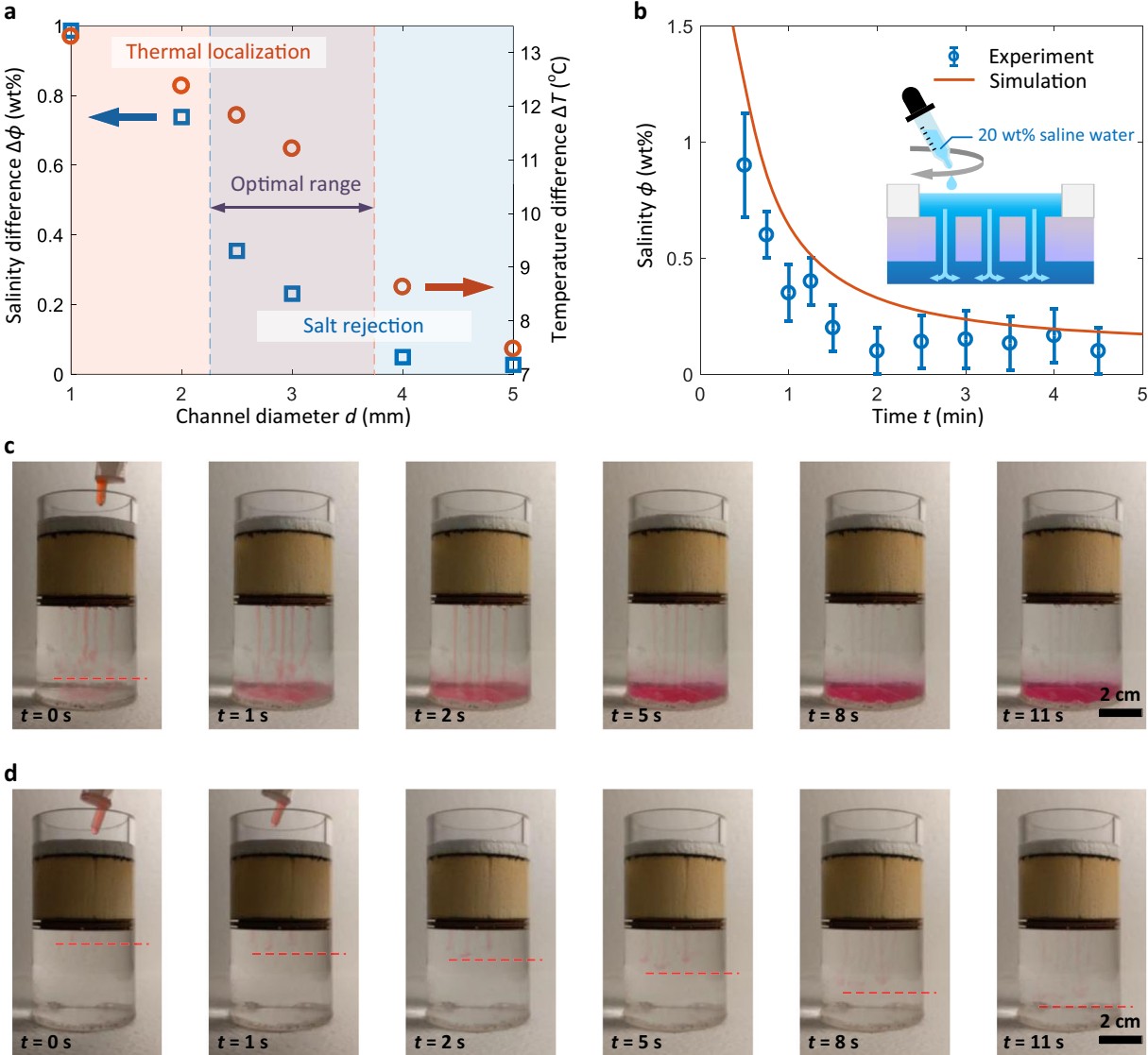

**Fig. 3 Fluidic flow engineered in the confined water layer. a** Salinity ($\Delta\phi$) and temperature ($\Delta T$) difference across the macrochannel as a function of the macrochannel diameter $d$. The sharp transition of $\Delta\phi$ and $\Delta T$ (marked by the blue and red dashed lines, respectively) indicates the convection-dominated regime. The overlap of thermal localization (light red regime) and salt rejection (blue regime) represents the optimal design for the macrochannel diameter. **b** Confined water layer salinity $\phi$ as a function of time $t$ after dripping 2.3 mL 20 wt% brine in an isothermal condition. $\phi$ was measured by a digital refractometer. Error bars represent the combination of instrumental uncertainty of digital refractometer (0.1 wt%) and the standard deviation of three-time measurements. Good agreement between the experiment and model prediction (red curve) is shown. **c** Time-lapse image of the transport of 20 wt% brine visualized by food dye. **d** Time-lapse image of the transport of DI water visualized by food dye. For both visualization tests, the reservoir initially contained DI water only. The faster transport of 20 wt% brine (**c**) than DI water (**d**) confirmed the existence of the natural convection effect.

however, there was a sudden drop of $\Delta T$, indicating the onset of convection-dominated heat loss. For the salt transport process, the salinity difference $\Delta\phi$ follows a similar trend to $\Delta T$ while the sharp transition due to the onset of convection-dominated salt rejection occurs earlier at $d = 2.5$ mm (the blue regime in Fig. 3a). Therefore, when 2.5 mm $< d <$ 3.5 mm, there is a regime for simultaneous thermal localization and salt rejection (the purple band in Fig. 3a), which agrees with the qualitative analysis shown in Fig. 1d. We experimentally validated this regime by evaporating 20 wt% brine on three confined water layer structures with 1 mm, 2.5 mm, and 5 mm diameter macrochannels (see Supplementary Figs. 6 and 7 for details). Significant salt crystallization and increased heat loss through macrochannels were observed on the 1 mm and 5 mm diameter macrochannel designs, respectively, both of which reduce the solar-to-vapor conversion efficiency (see Supplementary Fig. 7 for details). Guided by the modeling and

experimental results, we chose $d = 2.5$ mm as the macrochannel diameter for our prototype.

Although the diameter of macrochannels is the most important design parameter to achieve the simultaneous thermal localization and salt rejection, careful optimization of the confined water layer thickness, macrochannel arrangement, and thermal insulation thickness was also performed to provide complete design guidelines for the confined water layer structure. In particular, the confined water layer thickness strongly affects the uniformity of salt concentration. Nonuniform salt concentration is undesirable for practical applications because salt crystallization will always occur at the position with the highest concentration. With mechanistic modeling of evaporating 20 wt% brine under one sun illumination, we showed that very thin confined water layer (<3 mm) will lead to significant spatial nonuniformity of salt concentration (≈1 wt% salinity difference) due to insufficient

in-plane salt transport. Meanwhile, too thick of a confined water layer (>10 mm) will also increase the nonuniformity because of the reduced cross-plane salt transport (see Supplementary Fig. 8a for details). In addition, the thick confined water layer will increase heat loss due to the increase of sidewall area (see Supplementary Fig. 8b for details). For these reasons, a 5 mm thick confined water layer was used in our design to ensure an optimal uniformity of salt concentration and insignificant heat loss. Five macrochannels were fabricated where one of them was located at the center of the thermal insulation and the other four were 9 mm away from the center (see Supplementary Fig. 2). We chose this configuration to further improve the salt concentration uniformity in the confined water layer. With mechanistic modeling, we show that although macrochannel spacing plays a relatively insignificant role in salt transport, the 9 mm macrochannel spacing is desirable to enable the highest uniformity of salt concentration (see Supplementary Fig. 9 for details). The thickness of thermal insulation was also optimized, and a 25 mm thick insulation layer was chosen (see Supplementary Note 3 for details).

We first confirmed the enhanced salt rejection by natural convection. The experiment was in an isothermal condition to decouple the thermal effect. The reservoir initially contained deionized (DI) water only. We uniformly dripped 2.3 mL 20 wt% brine onto the confined water layer within 15 s (inset of Fig. 3b). The syringe used for the dripping test was placed less than 5 mm above the water–air interface to reduce the initial velocity of saline drops. We measured the salinity of the confined water layer using a digital refractometer (see "Methods" for details of salinity measurement). Figure 3b shows the temporal evolution of the confined water layer salinity $\phi$ from three measurements at each time. The rapid decrease of $\phi$ indicates a strong natural convection of the high-density brine through macrochannels, which agrees well with the model prediction (red curve in Fig. 3b, see "Methods" for details of dripping test modeling). We further visualized this natural convection effect using red food dye, where 1 mL food-dye-colored 20 wt% brine dripped onto the confined water layer within 6 s. The rapid development of the convective flow through the macrochannels was clearly captured (Fig. 3c and Supplementary Movie). As a comparison, we also dripped 1 mL food-dye-colored DI water onto the confined water layer. As shown in Fig. 3d, however, the evolution of the food-dye-colored water under the floating thermal insulation was slower because the natural convection effect was weaker without the presence of high-density brine. In addition, despite the same concentration of red food dye used for both the 20 wt% brine and DI water dripping, most of the food-dye-colored DI water remained in the confined water layer throughout the entire test because of the insignificant convective flow, leading to the much lighter colored liquid trajectories in Fig. 3d than that in Fig. 3c (Supplementary Movie). Overall, the optimized macrochannel dimension enables superior salt rejection capability from the confined layer by tailoring natural convection flow.

**Simultaneous thermal localization and salt rejection**. Next, we demonstrated the simultaneous thermal localization and salt rejection during solar evaporation (Fig. 4a). A solar simulator was used to provide uniform solar illumination (1000 W m$^{-2}$, one sun, see "Methods" for details of experimental setup). Thermal localization in the confined water layer was visualized by an IR camera (Fig. 4b). We also simulated the temperature and salinity profiles when evaporating 3.5 wt% brine (Fig. 4c, d). The simulated temperature profile (Fig. 4c) shows reasonable agreement with the IR image (Fig. 4b). Due to the natural convection effect, circulation formed inside each macrochannel, where the low-

salinity brine flowed upward to the confined water layer while the high-salinity brine flowed downward, creating a jetting region at the outlet of the macrochannel (Fig. 4d). Note that due to the careful design of the fluidic flow regime (Fig. 3a), no thermal circulations were observed in Fig. 4c. Therefore, heat transfer was still governed by conduction, resulting in a linear temperature gradient across the insulation layer. We performed more accurate temperature characterization of the confined water layer using a thermocouple (inset of Fig. 4e). The measured temperature response agrees well with the model prediction (red curve in Fig. 4e), where the confined water layer reached the thermal steady state after ≈30-min operation (light red regime in Fig. 4e). The steady-state temperature of the confined water layer was ≈40 °C (under one sun illumination), which is comparable with state-of-the-art wick structure-based evaporators[3,6,8,18,31,33–35].

To show the significant enhancement of salt rejection, we measured the temporal evolution of confined water layer salinity when evaporating 3.5 wt% (Fig. 4f), 10 wt% (Fig. 4g), and 20 wt% (Fig. 4h) brine under one sun illumination. As shown in Fig. 4f, the salinity of the confined water layer rapidly increased from 3.5 wt% to ≈4 wt% within the first hour. This result was because the evaporation rate increased with the confined water layer temperature (Fig. 4e), while the natural convection was not fully developed at the initial stage. A transition point occurred at $t = 1$ h, indicating a quasi-steady state of salt transport driven by fully developed natural convection (blue regime in Fig. 4f). The temporal evolution of salinity, especially the transition point, was well-predicted by the modeling results (red curve in Fig. 4f), demonstrating that the developed model captured well the underlying physics of salt transport. Note that we refer to the regime after the transition point as "quasi-steady" because there was still a slow increase of salinity with time (<0.5 wt% increase during 5 h). In our experiment, the amount of water in the reservoir continued decreasing due to evaporation, leading to a slow increase of salinity in the bulk water (yellow-dashed curve in Fig. 4f) and hence the quasi-steady state. Similar salt transport characteristics, i.e., a rapid build-up followed by a slow increase of salinity, were observed when evaporating 10 wt% (Fig. 4g) and 20 wt% (Fig. 4h) brine and well-captured by our developed model. No salt crystallization was observed in all experimental conditions because the highest salinity (23.3 wt%) after 6-h operation was lower than the saturated salinity (26.3 wt%). Note that for higher salinity brine, the transition to the quasi-steady state was faster (e.g., within a half hour for 20 wt%) due to the stronger natural convection induced by a larger density (salinity) gradient (Fig. 4h). This feature highlights the superior performance of salt rejection driven by natural convection. Therefore, compared with previous confined water layer structures[17,24], our design is capable of evaporating highly concentrated brine (20 wt%) without salt crystallization.

**Laboratory test**. We characterized the solar evaporation performance in a laboratory environment. Figure 5a shows a schematic of the experimental setup. The sidewall of the confined water layer and insulation layer were surrounded by thermal insulation ≈5 cm thick (polystyrene foam), which was covered by a double-layer aluminum foil to eliminate solar heating. A digital balance was used to measure the mass loss of the reservoir continuously. Three thermocouples were used to measure the real-time temperature of the confined water layer, bulk water, and ambient air, respectively. The thermocouple for the bulk water measurement was placed ≈2 cm below the floating insulation (Fig. 5a). The temperature and mass loss data were collected by data acquisition equipment and processed by a computer (PC, see "Methods" for details of experimental setup). To show the broad applicability of

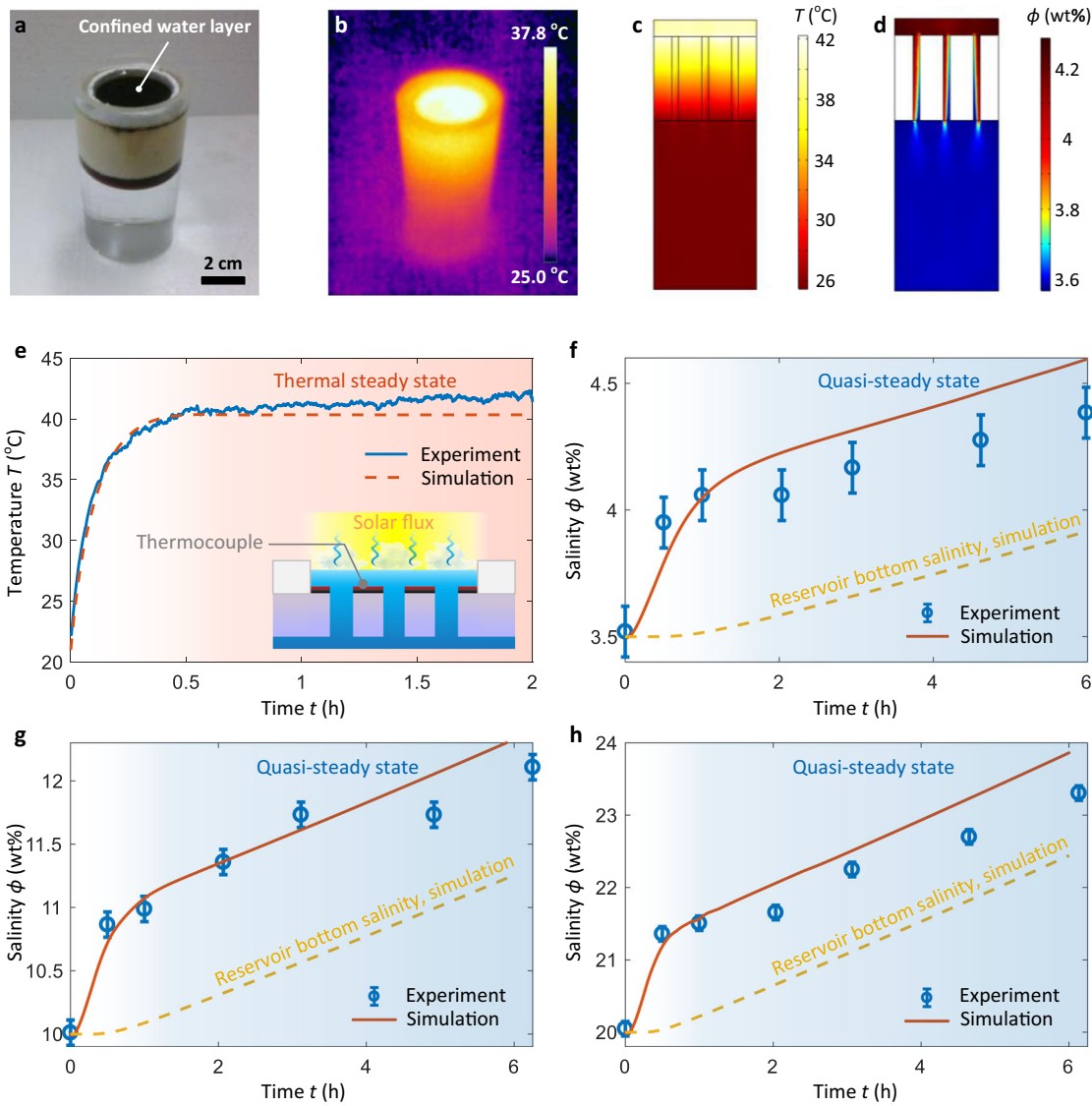

**Fig. 4 Simultaneous thermal localization and salt rejection during solar evaporation.** Photograph (**a**) and IR image (**b**) of the confined water layer structure in a reservoir. Thermal localization was confirmed by the IR characterization. Corresponding simulated temperature (**c**) and salinity (**d**) profiles for the evaporation of 3.5 wt% brine under one sun illumination. Circulations of salt inside each macrochannel due to natural convection are shown in the salinity distribution, while there remains a linear temperature gradient across the insulation layer. **e** Temperature response of the confined water layer under one sun illumination. The thermocouple measured temperature shows good agreement with model prediction. **f–h** Salinity response of the confined water layer for evaporation of 3.5 wt% (**f**), 10 wt% (**g**), and 20 wt% (**h**) brine under one sun illumination. Error bars represent the instrument uncertainty of the digital refractometer (0.1 wt%). The salinity of the confined water layer shows a rapid increase followed by a quasi-steady state, which is well-captured by the model prediction (red curve). Yellow-dashed curves represent the predicted increase of salinity in the bottom of the reservoir due to continuous evaporation, which leads to the salinity increase in the quasi-steady state.

the confined water layer structure, we tested three configurations. Configurations 1 and 2 were both typically used confined water layer evaporators without and with a convection cover, respectively (Fig. 5b). The convection cover consisted of two 45 mm diameter glass slides (2 mm thickness) separated by a 5 mm air gap. Configuration 3 was a contactless mode confined water layer evaporator, where both sides of the solar absorber were painted black (Fig. 5b). The same convection cover was placed on the top of the solar absorber to suppress the convective and radiative heat losses from the absorber. The confined water layer structure used for configuration 3 is the same as that used for configurations 1 and 2, with the only difference of removing the black paint from the top of the floating insulation (Fig. 5b).

Figure 5c shows the temperature response of configuration 1 during a 2-h operation. In the thermal steady state (light red

regime of Fig. 5c), the temperature of the confined water layer (blue curve in Fig. 5c) was more than 10 °C higher than the ambient temperature (yellow curve in Fig. 5c). Due to the strong thermal localization, the temperature of bulk water (red curve in Fig. 5c) only increased slightly and became close to the ambient temperature. A similar temperature response can be seen in configuration 2, where the temperature difference between the confined water layer and the ambient temperature was ≈15 °C (see Supplementary Fig. 10). Note that due to the small convective heat loss in the laboratory environment, the improvement of thermal localization with a convection cover was not significant. However, the convection cover would be necessary for outdoor operation where wind can significantly increase convective heat loss. Figure 5d shows the temperature response of configuration 3. The temperature of the solar absorber rapidly

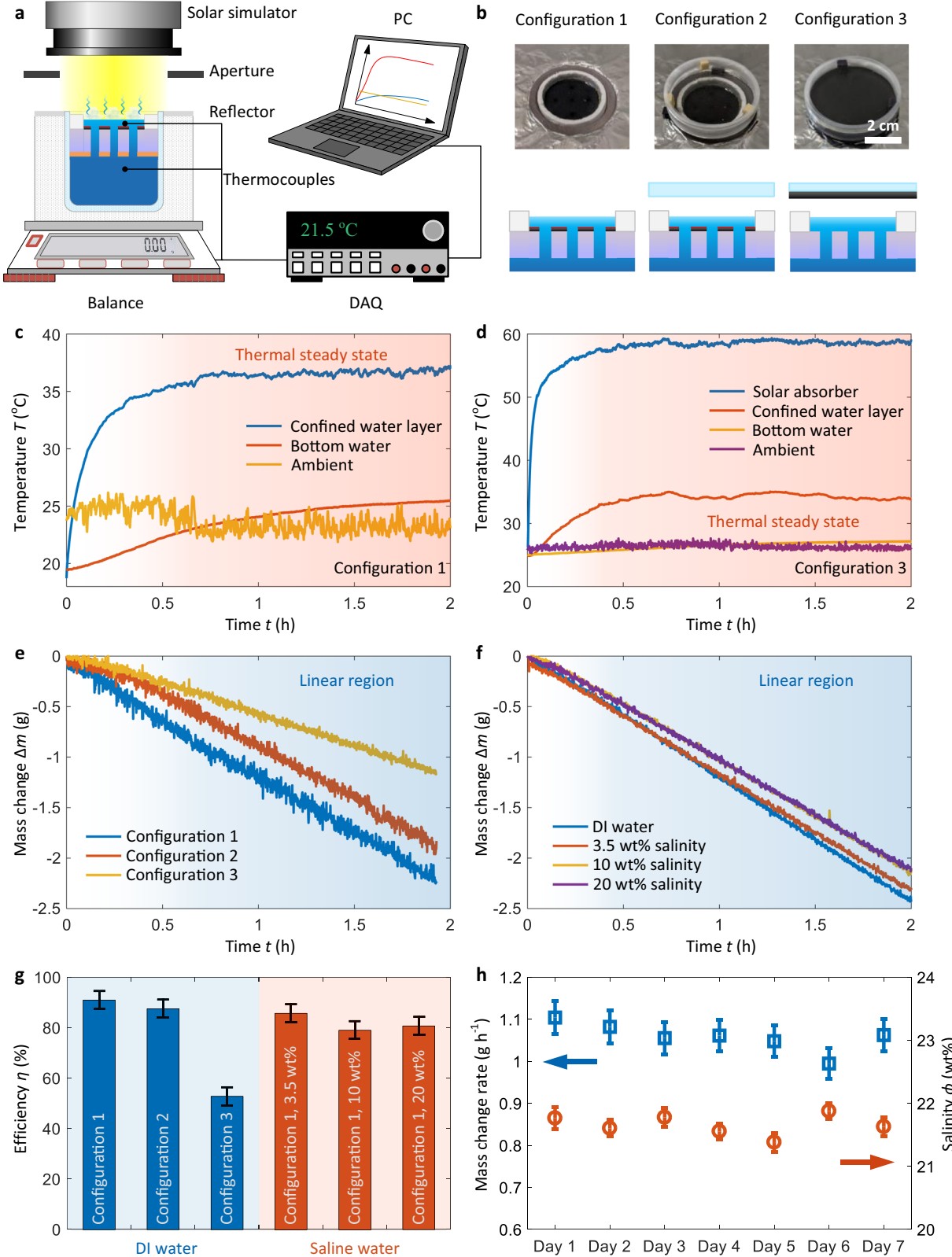

increased to 50 °C within ≈5 min and reached as high as ≈60 °C in the thermal steady state (blue curve in Fig. 5d). The confined water layer was heated by IR thermal radiation of the solar absorber. The temperature of confined water layer (red curve in Fig. 5d) was ≈10 °C higher than the bulk water temperature (yellow curve in Fig. 5d) and ambient temperature (purple curve in Fig. 5d). Compared with configurations 1 and 2, the solar absorber of configuration 3 has much higher temperature (60 °C). Therefore, vapor on the water–air interface of configuration 3 can be reheated by the solar absorber, which is promising for high-temperature steam generation.

Figure 5e shows the mass change of the reservoir during DI water evaporation. The rate of mass change increased gradually and reached a constant value in the thermal steady

**Fig. 5 Solar evaporation performance in a laboratory environment. a** Experimental setup consisting of a solar simulator, an aperture, a digital balance, a DAQ, and a PC. The reservoir containing the confined water layer structure was surrounded by thermal insulation. Aluminum foil covered the thermal insulation to serve as a solar reflector. The aperture and solar reflector ensured only the confined water layer as heated by the solar energy. **b** Three configurations used for solar evaporation experiment. Configurations 1 and 2 were normal mode evaporators where the solar-thermal conversion occurred at the bottom of the confined water layer. A convection cover was installed in configuration 2. Configuration 3 represents the contactless mode evaporator where the solar absorber is separated from the water surface by an air gap. Top panel: images of three configurations. Bottom panel: side-view schematics of three configurations. **c** Temperature response of configuration 1 under one sun illumination. The light red regime represents the thermal steady state. **d** Temperature response of configuration 3 under one sun illumination. **e** Mass change of three configurations as a function of time for the evaporation of DI water under one sun illumination. **f** Mass change of configuration 1 as a function of time for the evaporation of DI water, 3.5 wt%, 10 wt%, and 20 wt% brine under one sun illumination. **g** Solar-to-vapor conversion efficiency for different configurations and salinities. Error bars were determined by the uncertainties of linear fitting and evaporator size. **h** Reliability test over a week with 8-h continuous evaporation of 20 wt% brine each day. Mass change rate includes the contribution of both solar evaporation and dark evaporation. Error bars of the mass change rate were determined by the uncertainties of linear fitting and evaporator size. Error bars of the salinity represent the combination of the instrument uncertainty of the digital refractometer (0.1 wt%) and the standard deviation of six-time measurements.

state. Compared with configuration 1, configurations 2 and 3 took longer time to reach the steady state due to the additional thermal mass of the convection cover. The evaporation rate was determined by fitting the linear region of mass change curves and excluding the contribution of dark evaporation (see Supplementary Fig. 11 for details of dark evaporation characterization). The evaporation rate of configuration 1 was $\approx$1.36 L m$^{-2}$ h$^{-1}$, corresponding to a solar-to-vapor conversion efficiency $\eta$ of 91%, which is comparable with the high-performance wick structure-based solar evaporators[3,6,8,36–40]. Although a higher temperature of the confined water layer was achieved in configuration 2, its evaporation rate (1.3 L m$^{-2}$ h$^{-1}$, $\eta = 87$%) was slightly lower than configuration 1. The reduction in evaporation rate was attributed to the increased vapor transport resistance and optical loss from the convection cover. Owing to the improved thermal localization with the confined water layer structure, configuration 3 achieved a high evaporation rate (0.75 L m$^{-2}$ h$^{-1}$) and solar-to-vapor conversion efficiency (51%) for the contactless solar evaporation, which is higher than previous studies[16,32].

We further demonstrate that the high evaporation rate can be maintained during brine evaporation. Figure 5f shows the mass change of the reservoir with different salinities (3.5 wt%, 10 wt%, and 20 wt%). We used configuration 1 for all of the tests, because it has the same confined water layer structure for salt rejection. No salt crystallization was observed during the tests. With the increase of salinity, there was a decrease in evaporation rate mainly due to the reduced water activity of salt solutions[9]. However, for the 20 wt% brine, the solar-to-vapor conversion efficiency was still higher than 80%. Moreover, to show the limit of the confined water layer structure, we performed 6-h continuous solar evaporation of 25 wt% brine, which is approaching the saturation level of NaCl in water at room temperature (26.3 wt%). No salt crystallization was observed during the 6-h test and 67% solar-to-vapor conversion efficiency was demonstrated (see Supplementary Fig. 12 for details). We summarized the solar-to-vapor conversion efficiencies for the three different configurations and salinities in Fig. 5g. The superior performance was further confirmed by comparing our prototype with representative solar evaporators reported in recent studies (see Supplementary Table 1 for details).

To examine the reliability of the confined water layer structure, we performed solar evaporation of 20 wt% brine over a week. For each day, the solar evaporation experiment continued for 8 h under one sun illumination. The total solar irradiation was 8 kW h m$^{-2}$, which is much larger than the US annual average daily solar irradiation ($\approx$4.5 kW h m$^{-2}$)[41]. After the 8-h test, the solar simulator was turned off for 16 h to emulate nighttime conditions. The steady-state rate of mass change and the salinity of the confined water layer were measured (see "Methods" for

details of the reliability test). No fouling due to salt crystallization occurred. As shown in Fig. 5h, the mass change rate and salinity were stable after 1-week operation, indicating reliable performance of the confined water layer structure.

**Outdoor test**. To further understand the performance of the confined water layer structure in realistic weather conditions, we conducted an outdoor experiment on a sunny day (October 14, 2020). Figure 6a shows two identical experimental setups placed next to each other in East Setauket, New York, USA (see Supplementary Note 4 for details of the outdoor test): one contained a normal mode evaporator with a convection cover (configuration 2, right-hand side of Fig. 6a) while the other had a contactless mode evaporator (configuration 3, left-hand side of Fig. 6a). Since convection due to natural wind was significant in the outdoor condition, the convection cover became necessary to suppress heat losses. The incident solar flux was measured by a pyranometer (see Fig. 6a and "Methods" for details). The experiment started at 10:30 (local time) and ended at 15:30 (local time). Figure 6b shows the solar flux, confined water layer temperature of configuration 2, and ambient temperature as a function of time. The convection cover installed in configuration 2 was used to suppress the convective heat loss due to wind. The average solar flux during the 5-h operation was 595 W m$^{-2}$. The confined water layer had a similar temperature response to the laboratory characterization. In the thermal steady state of configuration 2, the temperature difference between the confined water layer and ambient air was larger than 10 °C, indicating comparable thermal localization to the laboratory condition. The temperature response of configuration 3 was recorded in Fig. 6c. The peak temperature of solar absorber reached $\approx$50 °C, leading to a $\approx$10 °C increase of the confined water layer temperature above the ambient temperature. Figure 6d shows the mass change of the two configurations due to evaporation. The significant fluctuations in the data arose from the time-varying wind load acting on the setups. We determined the evaporation rate by linearly fitting the mass change data. The evaporation rates of configurations 2 and 3 were 0.91 L m$^{-2}$ h$^{-1}$ and 0.55 L m$^{-2}$ h$^{-1}$, respectively (including the contribution of dark evaporation), which were comparable with their laboratory performance.

## Discussion
This experimental demonstration of the wick-free confined water layer structure provides a simple and low-cost approach to achieve highly efficient and salt rejecting solar evaporation. Our approach takes advantage of a moderate amount of water confinement to relax the inherent tradeoff between the thermal localization and salt rejection in previous wick structure-based

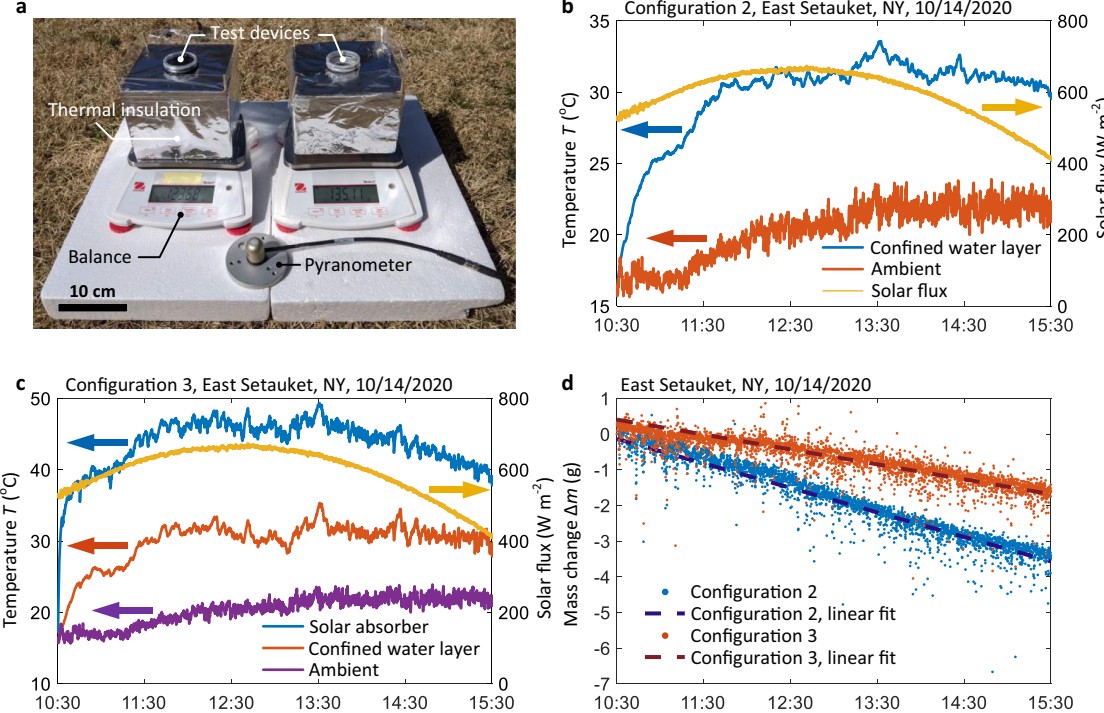

**Fig. 6 Outdoor test of the confined water layer structure in East Setauket, NY, USA. a** Image of the experimental setup comprising of two test devices, thermal insulation, two digital balances, and a pyranometer. Two identical setups were placed next to each one, containing configuration 2 (the right one) and configuration 3 (the left one), respectively. **b** Real-time variations of solar flux, confined water layer temperature (configuration 2), and ambient temperature during the outdoor test. **c** Real-time variations of solar flux, contactless solar absorber temperature (configuration 3), confined water layer temperature (configuration 3), and ambient temperature during the outdoor test. **d** Real-time mass change variations of configurations 2 and 3 during the outdoor test. The evaporation rates were determined from the slopes of the dashed lines using linear fitting.

and contactless solar evaporators. The self-floating feature also decouples the functionalities of solar-thermal conversion, thermal localization, and passive water supply, which further relaxes material constraints. Using commonly available materials, we demonstrate a prototype with solar-to-vapor conversion efficiency comparable to wick structure-based evaporators and salt rejection performance as good as contactless evaporators.

We believe that engineering passive fluidic flow is a promising while not fully explored avenue toward significant enhancement of salt rejection. The fundamental understanding of salt transport plays a central role in manipulating the fluidic flow. This work develops a mechanistic model by coupling the salt transport with the fluidic flow and heat transport to quantitatively guide the evaporator design. We show that the natural convection due to the salinity gradient can be passively triggered by carefully engineering the macrochannels in the thermal insulation. More importantly, owing to the two orders of magnitude difference between the mass diffusivity of salt in water and thermal diffusivity of water, we theoretically and experimentally identified a regime where the convective flow significantly drives salt rejection while inducing negligible additional heat losses—the key to achieve simultaneous thermal localization and salt rejection. We believe this mechanistic model-driven, fully quantitative design approach can serve as general guidelines to interface fluidic flow engineering with various solar evaporation devices. In addition, it is possible to improve the resistance to biofouling by taking advantage of the convective flow[42–44], which requires further investigation in future works.

This work can be widely integrated into existing passive solar evaporators. We not only show the reliable performance of the confined water layer structure in the normal mode evaporators, but also extend it into the contactless mode and report an

improved solar-to-vapor conversion efficiency. With an optimized design, the developed confined water layer structure could further improve the reliability of passive solar desalination technologies and promote zero-liquid discharge in wastewater treatment.

## Methods

**Design and fabrication of the confined water layer prototype**. A circular polyurethane foam (36 mm diameter and 25 mm thickness) was used as the floating thermal insulation. An insulating ring (36 mm external diameter, 31 mm internal diameter, and 6 mm height) made of polystyrene foam was attached on top of the floating thermal insulation. Black paint (245198, Rust-Oleum) was uniformly sprayed on the top of the thermal insulation layer, creating a 31 mm diameter area for solar absorption. Five 2.5 mm diameter macrochannels were drilled through the thermal insulation using waterjet. One of five macrochannels was in the center of the floating thermal insulation, while the other four were in four vertices of a square, 9 mm away from the central macrochannel. A circular copper plate (36 mm diameter) was used as the balancing weight, which was attached to the bottom of the floating thermal insulation. Similar to the floating thermal insulation, five 2.5 mm diameter macrochannels were also machined through the copper plate using waterjet. The total weight of the copper plate was 23.4 g to enable the neutral buoyancy of the entire structure. The convection cover comprised two glass slides (45 mm diameter and 2 mm thickness) and an air gap (5 mm thickness). The solar absorber for the contactless mode was a double-sided black-painted aluminum plate, attaching to the back side of the convection cover.

**Optical property measurement**. The direct-hemispherical reflectance ($R$) of the solar absorber (from 250 nm to 2.5 μm wavelength) was characterized using a UV-vis-NIR spectrophotometer (LAMBDA 1050, Perkin Elmer) with an integrating sphere (PMT, InGaAs). The absorptance of the solar absorber ($\alpha$) was obtained by the direct-hemispherical reflectance ($\alpha = 1 - R$). The spectra averaged absorptance of solar absorber was 95.3%. Note that the solar absorptance was characterized in a dry state without water layer on the top of the solar absorber. The presence of the water layer will lead to a weak reflection to visible light (≤2% of the visible light according to Fresnel's law) while enhanced absorption to IR light (>1000 nm wavelength). Considering this combined effect, 95.3% can be a reasonable estimation for actual solar absorption during practical operations, which was used in our design and analysis.

**Dynamic response simulation.** The dynamic response of the confined water layer was phenomenologically described by a second-order system for a damped oscillator,

$$m\frac{d^2y}{dt^2} + \text{sgn}\left(\frac{dy}{dt}\right) \cdot C_D \rho A_s \left(\frac{dy}{dt}\right)^2 = mg - \rho g V(y) \qquad (5)$$

where $m$, $\rho$, $A_s$, and $g$ are the total mass of the floating structure (solid part), water density, top surface area of the floating structure, and magnitude of gravitational acceleration, respectively. $V$ is the total immersed volume of the floating structure (solid part). Note that $V$ is a constant value when the structure is fully immersed into water, while it becomes a function of the displacement $y$ when the top of the floating structure (the insulation ring) is above the water–air interface. A sign function sgn was used to ensure that the drag force acts along the opposite direction of the floating structure motion. $C_D$ is the drag coefficient determined by fitting. When the entire structure is immersed into water, $C_D$ remains a constant ($C_D = 2$). When the top of the floating structure (the insulating ring) moved above the water–air interface, an additional drag term that linearly related to the displacement was added to capture the contribution due to surface tension. This model can qualitatively capture the dynamic behavior of the floating structure, where two regimes, i.e., the floating process dominated by buoyancy and the damped oscillation governed by both the buoyancy and drag, were shown.

**Salt transport modeling.** A mechanistic model of salt transport was developed in COMSOL Multiphysics 5.5. A 3D computational domain was constructed based on the geometry of the experimental prototype. The entire computational domain was resolved by ≈386000 nodes with the smallest node size of 0.3 mm. Mesh dependence analysis was performed (see Supplementary Fig. 5) to ensure the numerical accuracy. Equations (1–4) were solved simultaneously using a time-dependent solver. For heat transport, a uniform heat flux boundary condition was applied on the top of the floating thermal insulation to describe the solar-thermal heating. An evaporative heat flux boundary condition was applied to the water–air interface, which was determined by a heat transfer coefficient $h_{evap} = 53\ \text{W m}^{-2}\text{K}^{-1}$ and the corresponding temperature rise. This evaporative heat transfer coefficient was calibrated from experiments. An additional heat transfer coefficient $h_{natutral} = 5\ \text{W m}^{-2}\text{K}^{-1}$ was also applied to the water–air interface to quantify the heat loss due to natural convection[45]. For salt transport, a mass flux boundary condition was applied to the water–air interface to model the salt accumulation due to evaporation. The mass flux was given by the evaporative flux and bulk salinity. The higher evaporation rate or larger bulk salinity leads to more significant salt accumulation. No flux boundary condition was applied to all of the other boundaries. More detailed descriptions, validations, and simulation results of the salt transport model can be seen in Supplementary Note 3 and Supplementary Fig. 4.

To model the dripping process of brine, a uniform layer of 20 wt% brine (2.3 mL) was created on the top of the confined water layer as the initial condition. Since solar evaporation was disabled in the dripping test, the heat transport module was not incorporated into the model. No flux boundary condition was applied to the water–air interface of the confined water layer for salt transport. In the experiment, the dripping process took 15 s. Therefore, the simulation time $t = 0$ s represents the real-time $t = 15$ s. The simulation time was converted to real time for the comparison of Fig. 3b.

**Experimental setup.** One sun solar flux was provided by a solar simulator (92192, Newport Oriel Inc.) in the laboratory condition. The beam size was controlled by an aperture. The reservoir was surrounded by a 5 cm thick thermal insulation. Aluminum foil was used as the solar reflector, which was covered on the thermal insulation to ensure that only the confined water layer was heated by solar flux. Mass loss of the reservoir was measured by a digital balance (SJX6201N/E, Ohaus). Temperatures were measured using an IR camera (FLIR C5) and K-type thermocouples (Omega 5TC-TT-K-36-36). Mass loss and temperatures were recorded by a DAQ (34972 A, Agilent) and processed by a PC. The total rate of mass loss $\dot{m}_{tot}$ was obtained from the linear fitting of the mass loss curve in steady state, where $\dot{m}_{tot}$ is equal to the summation of solar evaporation rate $\dot{m}$ and dark evaporation rate $\dot{m}_{dark}$. The contribution of dark evaporation for the three configurations was calibrated in the same laboratory condition without solar illumination (see Supplementary Fig. 7 for details of dark evaporation characterizations). $\dot{m}$ was thus determined by carefully excluding $\dot{m}_{dark}$ from $\dot{m}_{tot}$ and used to determine the solar-to-thermal conversion efficiency $\eta$,

$$\eta = \frac{\dot{m} h_{fg}}{A q''_{solar}} \qquad (6)$$

where $h_{fg}$ is the vaporization enthalpy, $q''_{solar}$ is incident solar flux, and $A$ is the solar absorbing area. The salinity was measured using a digital refractometer (HI 96801, Hanna Instruments). 100 μL brine was carefully collected by a pipette (VWR High Performance Single-Channel Pipettors) from the water–air interface and then dispensed into the stainless steel well of the digital refractometer. For the outdoor test, the incident solar flux was measured by a pyranometer (SP-510-SS, Apogee).

**Reliability test.** Cycle tests for reliability were performed over a week (7 cycles in total). Each cycle consists of 8-h continuous evaporation under one sun illumination (to simulate the daytime operation) and 16-h dark process without solar illumination (to simulate the nighttime operation). 20 wt% brine was used for the reliability test. In each cycle, the mass change rate was determined from the steady state of the first 2-h operation. At the end of the second hour, DI water was slowly infused from the bottom of the reservoir (1 mL h$^{-1}$) by a syringe pump (PHD ULTRA 4400 Programmable Syringe Pump, Harvard Apparatus). We infused water to avoid the saturation of bulk salinity (see Fig. 4h) and compensate the loss of water due to continuous operation. Therefore, the mass transport could reach a fully steady state rather than the quasi-steady state in Fig. 4. Since the water was supplied from the bottom of water reservoir at a slow rate, its impact on the top confined water layer was minimized. The salinity of the confined water layer was measured at the eighth hour of the evaporation process. After the 8-h evaporation, both the solar simulator and syringe pump were turned off. No water was replaced during the entire dark evaporation process. Both the mass loss rate and salinity remained stable throughout the test. The average values of mass loss rate and salinity were 1.06 g h$^{-1}$ and 21.7 wt%, respectively. The mass change rate showed good agreement with the results in Fig. 5f and the salinity matched the transition point in Fig. 4h, indicating a reliable performance of the confined water layer structure.

## Data availability
All relevant data will be made available upon reasonable request from the authors.

## Code availability
The codes used in this work are available upon reasonable request from the authors.

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

## Acknowledgements

L. Zhang and E.N.W. acknowledge support from the Singapore-MIT Alliance for Research and Technology (SMART) LEES Program. The authors would like to thank Dr. Mian Wang at Stony Brook University for the help of the outdoor test setup. This work made use of the MRSEC Shared Experimental Facilities at MIT, supported by the National Science Foundation under award number DMR-1419807. A.L. acknowledges support from the US-Egypt Science and Technology Joint Fund. This article is derived from the Subject Data funded in part by NAS and USAID, and that any opinions, findings, conclusions, or recommendations expressed in such article are those of the authors alone, and do not necessarily reflect the views of USAID or NAS.

## Author contributions

L. Zhang and X.L. contributed equally to this work. L. Zhang, X.L., and Z.X. conceived the initial concept. X.L., L. Zhang, and L. Zhao developed the experimental setup and performed experimental characterization. X.L., L. Zhang, and A.L. developed the theoretical model and performed numerical simulation. L. Zhang, X.L., and Z.X. interpreted the theoretical and experimental results. Y.Z. performed material characterization. L. Zhang and X.L. wrote the manuscript with input from all authors. E.N.W. supervised and guided the project.

## Competing interests

The authors declare no competing interests.
