## [Peer Review File · Nature Communications]

REVIEWER COMMENTS

Reviewer #1 (Remarks to the Author):

This work reports highly efficient ($\approx 90\%$ solar-to-vapor conversion efficiency) and salt rejecting (20 weight % salinity) solar evaporation by engineering the fluidic flow in a confined water layer. They show that natural convection can be used in the confined water. There exists a regime enabling simultaneous thermal localization and salt rejection, i.e., natural convection significantly accelerates salt rejection while inducing negligible additional heat loss. Furthermore, they show the broad applicability by integrating this confined water layer with a recently developed contactless solar evaporator and report an improved efficiency. As such, the subject matter is of interest. However, the paper suffers for severe limits:

1. The findings that have presented regarding simultaneous thermal localization and salt rejection are interesting. But this concept is not a strong candidate for the very broad readership and real applications. There would need to be a greater conceptual advance.
2. Solar interfacial evaporation via near-field radiation to prevent salt blocking is interesting but the design rules or guides need to be further studied in order to give a whole picture of this work. Besides, they should offer their opinions and comments on the physical novelty of this evaporation method.
3. They should compare their results with other related works for salt rejecting evaporation to highlight their advantages and disadvantages.

Reviewer #2 (Remarks to the Author):

Natural and passive convection is probably the most widely applied approach to realize practical implementation of salt-rejecting solar desalination, but previous studies rarely provide useful in-depth exploration to dissect this process, thus I find encouraging findings regarding this point in the current manuscript by Zhang et.al. Unlike most of previously reported solar evaporators, the developed solar absorber is completely immersed below water level, but still capable of realizing a very high conversion efficiency ($>90\%$), which is an interesting and surprising finding and worthy of such a study. This novel and effective salt rejection strategy presented in this work is also of significance to facilitate understanding of the topic of sustainable solar desalination. I thus think it is highly suitable for publication in Nature Communications, but some key issues should be carefully addressed, such that its quality could get further improved, specifically as follows:

1. The developed solar-evaporation configuration works in a way quite different from traditional solar basin (Adv. Sci. 2020, 7, 1903478.), as the black liner of the current device is much closer to

water level, and of course, with proper insulation beneath it. Thus, the total volume of salt water within this confined space is important to suppress volumetric heating and following conduction loss, and at the same time, enable sufficient mass exchange, i.e., the convection, to avoid salt formation. Then the thickness of the confined water layer is considered a key parameter to this trade-off, but this critical investigation was not provided here. How was it determined in the current manuscript? and by changing it, how will it influence the performance? Although the conclusion is obvious that an optimal thickness is attainable under a given configuration, these questions should be carefully discussed.

2. The solar absorbance is 0.953 as reported by the authors, I suppose it was measured in completely dry state, since in the actual scenario, a confined water layer is covering the solar absorber, and with this reflective water film, the overall absorbance could not be so high. In the simulations, 0.953 was taken into account as well, if not considering this likely reduction in solar absorption caused by the water film, the calculated outcome might not be accurate any longer.

3. With this overestimated solar absorbance, which requires further confirmation by the authors, the overall solar efficiency was 91% as reported by the authors, which is surprisingly high, especially considering that the solar absorber was immersed beneath the water level instead of at water/air interface. Thus, heat losses due to different outlets such as conduction, radiation, etc. that could account for this 91% must be specified and analyzed.

4. Clarifications are needed regarding the demonstrations shown in Fig. 3C and 3D: 1) this dripping action for the two tests should be identically synchronous for a strict comparison, but there was still tracer liquid left in the syringe at 1s (3C) while the injection in 3D was completed already; 2) It is not clear what the inset caption (DI water and Brine) refers to (the bulk water or the dyed water?); 3) it is hard to tell how convection effect makes a difference as the red fluid reached to the bottom slowly in 3C, while the other one (3D) was obviously faster but gathered, besides, the shade of dyed water (one is light, one is dark) is also a problem to make a direct and clear comparison.

5. A particular concern regarding the immersed solar absorber is the bio-fouling issue. It is not like other capillary driven systems where bio-impurities could be rejected simply by spontaneous capillary filtration, such as the unidirectional transportation (as reported in ACS Energy Lett. 2020, 5, 11, 3397–3404; Desalination. 2021, 515, 115192; ACS Appl. Mater. Interfaces 2021, 13, 32, 38405–38415), how can the current device prevent itself from bio-fouling?

6. Others: 1) information like the model numbers and brands of instruments employed in the work are found twice in the main text and in the experimental section, which seems of no need, it would be better to just leave them in the experimental section; 2) same symbol was used for salinity and diameter (Fig.S2); 3) the unit of salinity is not consistent (wt% in the main text, w% in SI).

Reviewer #3 (Remarks to the Author):

In this article, the authors reported a solar evaporation system based on the employment of wick-free water transportation pathway and a confined water layer. The key point and advance are the realization of thermal localization and salt rejection convection from the view point of fluidic flow

optimization. The article is very well written and organized. However, I still have some reservations, on addressing which the article can be improved further.

Detailed comments:

1. The authors emphasized the use of the wick-free confined water layer, as salt accumulation phenomenon, especially when evaporating high salinity brine, cannot be fully addressed by the wick-based system. However, it is reasonable to think of a wick-based system with similar arrayed channel configuration and fluidic regulation (Adv. Mater. 2019, 31, 1900498, Reference 16 in this manuscript) where similar or even better performance can be achieved (1.46 kg m⁻² h⁻¹ evaporation rate for DI water, 100 hours continuous salt accumulation free, and 60 days stability, etc.), comparing to this system. The advantage of this wick-free system over the wick-based system seemed not to be clear enough with current data and analyses. Further investigation and optimization are needed to satisfy the high standard of Nature Communications.
2. As described in Line 381, Page 17, 91% is the efficiency for DI water. Is the efficiency for 20 wt% NaCl solution also 91%? If not, the statement of “Here, we demonstrate highly efficient (\approx 90% solar-to-vapor conversion efficiency) and salt rejecting (20 weight % salinity) solar evaporation by engineering the fluidic flow in a wick-free confined water layer” in abstract should be revised, otherwise it will lead to misunderstanding.
3. The optimization of the macrochannel dimension and the insulation thickness was acquired through simulation only. Experimental results are suggested with simulation results as assist.
4. Why did the authors choose such arrangement and number of the macrochannels in this manuscript?
5. Does the wettability of the water pathway, i.e., the macrochannel in this system, influence the water transportation or confined water thickness, and further the performance of the solar evaporation system? In addition, the wettability of the polyurethane foam should be provided.
6. The thickness of the confined water layer, \sim 5 mm in this work, should be regulated and investigated as the water layer bottom is directly heated while evaporation occurs on its surface.
7. The locations of Fig. 3c and Fig. 3d are inconsistent with the description in the main text and the figure captions.

Response Letter

Reviewer 1

This work reports highly efficient ($\approx 90\%$ solar-to-vapor conversion efficiency) and salt rejecting (20 weight % salinity) solar evaporation by engineering the fluidic flow in a confined water layer. They show that natural convection can be used in the confined water. There exists a regime enabling simultaneous thermal localization and salt rejection, i.e., natural convection significantly accelerates salt rejection while inducing negligible additional heat loss. Furthermore, they show the broad applicability by integrating this confined water layer with a recently developed contactless solar evaporator and report an improved efficiency. As such, the subject matter is of interest. However, the paper suffers for severe limits:

Response: We really appreciate that the Reviewer found our work interesting and provided valuable comments to help us improve the quality of our work. We carefully considered all of comments and provide detailed responses as follows.

1. The findings that have presented regarding simultaneous thermal localization and salt rejection are interesting. But this concept is not a strong candidate for the very broad readership and real applications. There would need to be a greater conceptual advance.

Response: We agree with the Reviewer that it is very important to provide broader context for general readership. As suggested by the Reviewer, we highlighted that solar evaporation has been recognized as a promising means to address the increasingly severe global water shortage [1]. For this reason, enabling highly efficient and reliable solar evaporation is critical for real-world implementation of sustainable water treatment [2-5]. Since the overall energy efficiency mainly depends on thermal localization [2,3] and the reliability can be improved by the enhanced salt rejection [4,5], achieving simultaneous thermal localization and salt rejection has become one of key bottlenecks for high-performance solar evaporators [4,5]. We believe that the key conceptual advance demonstrated in this work is our ability to demonstrate simultaneous thermal localization and salt rejection. To make this point clear, we rewrote the introduction section in Page 3, Line 2 of the revised main text,

“Water scarcity has become a severe challenge for humanity since two-thirds of the global population is affected by water shortage¹. Owing to the significant potential for clean water production, highly efficient solar evaporation by localizing the solar-thermal conversion process near the evaporating interface has attracted tremendous research interest in passive vapor generation²⁻⁷, seawater desalination⁸⁻¹³, wastewater treatment¹⁴⁻¹⁸, and medical sterilization¹⁹. However, due to the ultralow diffusivity of salt in water ($\sim 10^{-9} \text{ m}^2/\text{s}$, as a reference in comparison to the diffusivity of vapor in air is $\sim 10^{-5} \text{ m}^2/\text{s}$), there is significant salt accumulation, which induces undesirable fouling, reduces evaporation rate, and degrades device reliability. This effect has become one of the key practical challenges for a range of applications^{8,20-26}.

To enable highly efficient and reliable solar evaporation, a key bottleneck is to achieve simultaneous thermal localization and salt rejection....

1. Mekonnen, M. M. & Hoekstra, A. Y. Four billion people facing severe water scarcity. *Sci. Adv.* 2, e1500323 (2016).”

2. Solar interfacial evaporation via near-field radiation to prevent salt blocking is interesting but the design rules or guides need to be further studied in order to give a whole picture of this work. Besides, they should offer their opinions and comments on the physical novelty of this evaporation method.

Response: We thank the Reviewer for finding the concept of contactless solar evaporation interesting. We agree with the Reviewer that it is important to provide general design rules and the whole picture of contactless evaporator. In fact, we would like to note that the concept of solar interfacial evaporation *via* near-field radiation, *i.e.*, contactless solar evaporation, was firstly developed in Refs. R6 and R7, where we cited these two important references as Ref. 16 and 32 in our original manuscript, respectively. The general design rules for contactless solar evaporation have been shown in Refs. R6 and R7.

Although contactless solar evaporation exhibits strong resistance to salt accumulation, it suffers from relatively low solar-to-vapor conversion efficiency ($\leq 43\%$) [6]. In this work, we integrated our wick-free confined water layer structure into the contactless solar evaporation, which is denoted as the “configuration 3” in Fig. R1 and our manuscript. This integration maintains the same salt rejection capability as the conventional contactless solar evaporation [6,7] while improving the solar-to-vapor conversion efficiency above 50% by reducing the parasitic heat loss through the bulk brine, which is one of the key bottlenecks of the previous contactless solar evaporation designs [6]. Since the contactless solar absorber is above the evaporating surface, which has no interference with the confined water layer structure and hence has negligible impacts on the salt transport (configuration 3 in Fig. R1). For this reason, we used almost the same design of the confined water layer structure (*i.e.*, the same confined water layer thickness, thermal insulation, and macrochannel dimensions) for all three configurations (Fig. R1). The only difference of configuration 3 is that we removed the solar absorbing black paint from the top of the floating insulation (Fig. R1). To make this point clear, we added the following description in Page 17, Line 20 of the revised main text,

“Configuration 3 was a contactless mode confined water layer evaporator, where both sides of the solar absorber were painted black (Fig. 5b). The same convection cover was placed on the top of the solar absorber to suppress the convective and radiative heat losses from the absorber. The confined water layer structure used for configuration 3 is the same as that used for configurations 1 and 2, with the only difference of removing the black paint from the top of the floating insulation (Fig. 5b).”

Fig. R1. A copy of Fig. 5b from the main text, which shows three configurations used for solar evaporation experiment. Configurations 1 and 2 are normal mode evaporators where the solar-thermal conversion occurred at the bottom of the confined water layer. A convection cover was installed in configuration 2. Configuration 3 represents the contactless mode evaporator where the solar absorber is separated from the water surface by an air gap. Top panel: images of three configurations. Bottom panel: side-view schematics of three configurations.

We also provided broad context for the contactless solar evaporation in our original main text, which described its working principle and physical novelty. In general, despite relatively low solar-to-vapor conversion efficiency ($\leq 43\%$ in previous studies [6,7]) compared with wick-structure based solar interfacial evaporation ($> 60\%$), contactless solar evaporation has better salt rejection performance and is promising for the high-temperature steam generation. For the convenience of the Reviewer, we copied our descriptions and comments for the contactless solar evaporation from Page 3, Line 21 of the revised main text,

“On the other hand, contactless solar evaporation, converting sunlight into infrared (IR) thermal radiation with a non-contact solar absorber, directly localizes heat on the bulk water interface and therefore completely eliminates the use of a wick structure (Fig. 1b)^{16,32}. Superior salt rejection has been recently demonstrated with high salinity brine (up to 25 weight % (wt%)). Furthermore, since the vapor temperature is not pinned by the liquid-vapor interface, contactless solar evaporation is capable of producing superheated steam, which is promising for high-temperature applications such as solar sterilization³². However, due to the increased heat loss through the non-confined bulk water (Fig. 1b), the highest reported solar-vapor conversion efficiency of contactless solar evaporation (43%) is lower than that of conventional wick structure-based devices ($> 60\%$)^{3,16,20}.”

In fact, the significance of the contactless solar evaporation can also be confirmed by our experiment. Fig. R2 shows the temperature response of configuration 1 (Fig. R2a) and configuration 3 (Fig. R2b) under one sun illumination, which can also be seen in Fig. 5c,d of the main text. Compared with configuration 1, the contactless solar absorber reached much higher

temperature ($\approx 60\text{ }^{\circ}\text{C}$). Therefore, vapor on the evaporating interface ($\approx 33\text{ }^{\circ}\text{C}$, red line of Fig. R2b) can be reheated by the high-temperature solar absorber, which is significant for applications requiring high-temperature steam generation. To further highlight this physical novelty of the contactless solar evaporation, we added the following discussion in Page 18, Line 8 of the revised main text,

“Figure 5d shows the temperature response of configuration 3. The temperature of the solar absorber rapidly increased to $50\text{ }^{\circ}\text{C}$ within \approx five minutes and reached as high as $\approx 60\text{ }^{\circ}\text{C}$ in the thermal steady state (blue curve in Fig. 5d). The confined water layer was heated by IR thermal radiation of the solar absorber. The temperature of confined water layer (red curve in Fig. 5d) was $\approx 10\text{ }^{\circ}\text{C}$ higher than the bulk water temperature (yellow curve in Fig. 5d) and ambient temperature (purple curve in Fig. 5d). Compared with configurations 1 and 2, the solar absorber of configuration 3 has much higher temperature ($60\text{ }^{\circ}\text{C}$). Therefore, vapor on the water-air interface of configuration 3 can be reheated by the solar absorber, which is promising for high-temperature steam generation.”

Fig. R2. A copy of Fig. 5c,d from the main text, which shows (a) temperature response of configuration 1 and (b) configuration 3 under one sun illumination. The light red regime represents the thermal steady state.

In addition to showing details of the contactless solar evaporation, we highly agree with the Reviewer on the importance of providing a whole picture of this work. In our original main text, we mainly discussed the design principles of macrochannel diameter in the confined water layer structure, which is the most important parameter to trigger natural convection and enable simultaneous thermal localization and salt rejection. To show a whole picture for the design of the confined water structure, in the revised manuscript, we performed additional experiments and simulations to demonstrate the impact of (1) macrochannel diameter, (2) confined water layer thickness, and (3) macrochannel spacing on the solar-to-vapor conversion efficiency and salt rejection performance. For the convenience of the Reviewer, we summarize the key results as follows.

(1) In the original main text, we demonstrated that there exists a regime where natural convection dominates the salt rejection while having a negligible impact on heat loss. With numerical simulations, we obtained an optimal range of the macrochannel diameter (approximately 2.5 – 3.5 mm) to reach this regime and chose 2.5 mm diameter for our prototype. Since enabling quantitative manipulation of natural convection is one of the key contributions of our work, in our revised manuscript, we provided experimental characterizations to support our simulation and validate this optimal regime. Briefly, we experimentally showed salt crystallization and increased heat loss that occurred on the 1 mm and 5 mm diameter macrochannel designs, respectively, leading to a regime of simultaneous thermal localization and salt rejection with the 2.5 mm diameter macrochannel design. In Page 10, Line 22 of the revised main text, we added the following descriptions for our experimental validation,

“We experimentally validated this regime by evaporating 20 wt% brine on three confined water layer structures with 1 mm, 2.5 mm, and 5 mm diameter macrochannels (see Supplementary Figure 6, 7 for details). Significant salt crystallization and increased heat loss through macrochannels were observed on the 1 mm and 5 mm diameter macrochannel designs, respectively, both of which reduce the solar-to-vapor conversion efficiency (see Supplementary Figure 7 for details). Guided by the modeling and experimental results, we chose $d = 2.5$ mm as the macrochannel diameter for our prototype.”

We also created two new Supplementary Figures in the revised Supplementary Information (Supplementary Figure 6 and 7, Page 12 and 13) to show the corresponding experimental details,

Supplementary Figure 6. Top view of the (a) floating ring, insulation foams with (b) 1 mm, (c) 2.5 mm, and (d) 5 mm macrochannel diameters used for experimental characterizations. Unit is in mm.

Supplementary Figure 7. Solar evaporation of 20 wt% brine under one sun illumination. Three confined water layer structures with 1 mm, 2.5 mm, and 5 mm diameter macrochannels were tested (see Supplementary Figure 6 for detailed dimensions). (a) Temperature response of the confined water layer with different macrochannel diameters. The steady state temperature decreases with macrochannel diameters due to the increased heat loss through the floating insulation, which validates our theoretical prediction in Fig. 3a of the main text. (b) Corresponding mass change of the evaporation setup as a function of time. Confined water layer structure with 2.5 mm diameter macrochannels shows the highest evaporation rate, which was chosen for our final design. The reduced evaporation rate of the 1 mm diameter macrochannel design is attributed to the salt crystallization and increased convective heat loss due to the elevated water layer temperature, whereas the reduced evaporation rate of the 5 mm diameter macrochannel design is induced by the increased conductive heat loss through macrochannels. Time-lapse images of the confined water layer structure with (c) 1 mm and (d) 5 mm diameter macrochannels during a 3-hour continuous solar evaporation. Salt crystallization was observed on the 1 mm diameter macrochannel design after the second hour, because natural convection enhanced salt rejection cannot be initiated with small macrochannel diameters. Scalebar: 2 cm.

(2) In the revised main text, we discussed the role of the confined water layer thickness in salt transport and solar evaporation to justify the 5 mm thick confined water layer used in our design. With mechanistic modeling, we show that although the confined water layer thickness has insignificant impact on the average salt concentration of the entire confined water layer, it strongly affects the uniformity of salt spatial distribution. Uniform salt concentration is highly important since salt crystallization always occurs at the position with the highest concentration. In general,

we show that a very thin confined water layer leads to highly nonuniform salt concentration with low concentration around the macrochannel and high concentration close to the external insulation ring (Fig. R3a). Since the salt transport resistance along the confined water layer $R_{s,\parallel}$ (in-plane direction) is inversely proportional to the confined water layer thickness t (i.e., $R_{s,\parallel} \sim 1/t$), reducing t increases $R_{s,\parallel}$ and hence leads to the highly nonuniform salt concentration (Fig. R3a). On the other hand, although increasing t is desirable for the in-plane salt transport, too thick confined water layer will increase the cross-plane transport resistance $R_{s,\perp}$ (i.e., $R_{s,\perp} \sim t$) and thus enlarge nonuniformity of salt concentration crossing the confined water layer. The large thickness of confined water layer also increases heat loss due to the increase of sidewall area. The 5 mm thick confined water layer used in our design enables the highest uniformity of salt concentration (Fig. R3b) and avoids significant heat loss.

Fig. R3. Salt concentration profiles in (a) 1 mm and (b) 5 mm thick confined water layers obtained from simulations. Macrochannel diameters for both cases are 2.5 mm and solar illumination is one sun. The salt concentration profiles were taken at ≈ 0.5 hours after the simulation started.

To elucidate the effect of the confined water layer thickness, we added the following discussion in Page 11, Line 1 of the revised main text,

“Although the diameter of macrochannels is the most important design parameter to achieve the simultaneous thermal localization and salt rejection, careful optimization of the confined water layer thickness, macrochannel arrangement, and thermal insulation thickness was also performed to provide complete design guidelines for the confined water layer structure. In particular, the confined water layer thickness strongly affects the uniformity of salt concentration. Nonuniform salt concentration is undesirable for practical applications because salt crystallization will always occur at the position with the highest concentration. With mechanistic modeling of evaporating 20 wt% brine under one sun illumination, we showed that very thin confined water layer (< 3 mm) will lead to significant spatial nonuniformity of salt concentration (≈ 1 wt% salinity difference) due to insufficient in-plane salt transport. Meanwhile, too thick of a confined water layer (> 10 mm) will also increase the nonuniformity because of the reduced cross-plane salt transport (see Supplementary Figure 8a for details). In addition, the thick confined water layer will increase heat loss due to the increase of sidewall area (see Supplementary Figure 8b for details). For these reasons, a 5 mm thick confined water layer was used in our design to ensure an optimal uniformity of salt concentration and insignificant heat loss.”

We also created a new Supplementary Figure (Supplementary Figure 8, Page 13) to support the discussion in the main text with more detailed results.

Supplementary Figure 8. Effects of the confined water layer thickness on (a) salt transport and (b) heat loss predicted by simulations. Solar evaporation of 20 wt% brine under one sun illumination was simulated. The macrochannel diameter is 2.5 mm. (a) Nonuniformity of salt concentration $\Delta\phi_{max}$ as a function of the confined water layer thickness at the transition point to the quasi-steady state (i.e., ≈ 0.5 hours after the simulation started). $\Delta\phi_{max}$ is the maximum value of salt concentration difference in the confined water layer, which is defined as the difference between the highest salt concentration and the lowest salt concentration. Insets: top view of salt concentration profiles with 1 mm and 5 mm thick confined water layers. Nonuniform salt concentration is undesirable because salt crystallization will first occur at the position with the highest salt concentration. $\Delta\phi_{max}$ first rapidly decreases and then slightly increases with the increase of the confined water layer thickness, leading to a minimum point approximately at 5 mm, which was selected for our design. The large nonuniformity of salt concentration occurs at a low confined water layer thickness (e.g., 1 mm), because the resistance of salt transport along the confined water layer (i.e., the lateral transport) is inversely proportional to its thickness, resulting in a low salt concentration near the macrochannel with a high salt concentration close to the external floating ring (see the inset of (a)). The slight increase in salt concentration nonuniformity at the large thickness (e.g., 10 mm) is attributed to the increased resistance of salt transport crossing the confined water layer (i.e., the vertical transport). (b) Heat loss through the sidewall of a floating ring as a function of the confined water thickness. Heat loss increases linearly due to the increase of sidewall area.

(3) We provided justifications for the choice of macrochannel arrangement (i.e., five macrochannels with a spacing of 9 mm). In Page 11, Line 16 of the revised main text, we added the following discussion,

“Five macrochannels were fabricated where one of them was located at the center of the thermal insulation and the other four were 9 mm away from the center (see Supplementary Figure 2). We chose this configuration to further improve the salt concentration uniformity in the confined water layer. With mechanistic modeling, we show that although macrochannel spacing plays a relatively

insignificant role in salt transport, the 9 mm macrochannel spacing is desirable to enable the highest uniformity of salt concentration (see Supplementary Figure 9 for details)."

To further support the discussion in the main text, we created a new Supplementary Figure (Supplementary Figure 9) in the revised Supplementary Information (Page 15),

Supplementary Figure 9. Effects of macrochannel spacing on the nonuniformity of salt concentration predicted by simulations. Solar evaporation of 20 wt% brine under one sun illumination was simulated. The macrochannel diameter is 2.5 mm and the confined water layer thickness is 5 mm. (a) Top view of three representative insulation foams with 4 mm, 9 mm, and 13 mm macrochannel spacing used for simulations. Unit is in mm. (b) Nonuniformity of salt concentration as a function of macrochannel spacing. In general, $\Delta\phi_{\max}$ weakly depends on the macrochannel spacing, but too large of a macrochannel spacing will lead to an increase of salt concentration nonuniformity. 9 mm spacing was chosen for our design to achieve the optimal macrochannel arrangement.

We hope the above revisions with additional discussion, experiments, and simulations provide general guidelines for the design of confined water layer structure and thus now better show the whole picture and novelty of our work.

3. They should compare their results with other related works for salt rejecting evaporation to highlight their advantages and disadvantages.

Response: We agree with the Reviewer that the performance of our design should be compared with related works. In the revised manuscript, we compared our confined water layer structure with several highly impactful studies, where most of them were published in recent three years. Detailed comparisons are shown in Supplementary Table 1 in the revised Supplementary Information (Page 6). We are one of the only works that demonstrates a low-cost ($< \$10 \text{ m}^{-2}$) highly efficiency ($> 80\%$) solar evaporator capable of continuously desalinating ($> 6 \text{ h}$) nearly saturated brine ($\geq 20 \text{ wt}\%$) without salt crystallization.

In Page 19, Line 16 of the main text, we added one sentence to refer to the performance comparison, “*The superior performance was further confirmed by comparing our prototype with representative solar evaporators reported in recent studies (see Supplementary Table 1 for details).*”

For the convenience of the Reviewer, we copied Supplementary Table 1 as below,

Supplementary Table 1: Comparison of solar-to-vapor conversion efficiency η , salt rejection capability, and cost of various solar evaporators

Solar evaporators	η (%)	Salinity (wt%)	Continuous testing time (h)	Salt crystallization	Cost
	86	3.5^a			
This work	81	20^a	6 h	No	Low
	67	25			
Self-assembled aluminum nanoparticles⁴	57	2.75	1 h	No	Medium
Femtosecond laser rendered metal panel⁵	67 ^b	3.5	1 h	No	High
Fabric wick-polystyrene based solar evaporator⁶	55	3.5	0.55 h	No	Low
3D printed biomimetic solar evaporator⁷	96	25	9 h	Yes	High
Marangoni flow-driven salt rejection⁸	$\frac{47^c}{130^d}$	20	7 h	No	Medium
Filter paper-CNTs based solar evaporator⁹	$\frac{81}{71}$	$\frac{3.5}{13}$	600 h ^e	Yes ^f	Low

Electrospun Janus solar evaporator ¹⁰	66	20	1 h	No	Medium
Natural wood ¹¹	75	20	100 h	No	Medium
Bimodal porous solar evaporator ¹²	57	15	7 h	No	Medium
Janus wood ¹³	82	20	8 h	Yes ^g	Medium
Water lily inspired solar evaporator ¹⁴	79	10	8 h	No	Medium
MOF-derived porous carbon nanoflake arrays ¹⁵	91	10	2.5 h	No	High

^a7 cycles with 8 h per cycle; ^bclean water evaporation efficiency; ^csingle-stage operation; ^dthree-stage operation; ^e3.5 wt% saline water test; ^fsalt crystallization at around 0.5 h for 3.5 wt% saline water test; ^gsalt crystallization after 10-cycle operation.

1. *Polyurethane Foam Prices*. Available at: https://www.alibaba.com/product-detail/Hard-Surface-custom-high-quality-PVC_60456172600.html.
2. *Black Marine Coating Prices*. Available at: https://www.alibaba.com/product-detail/High-build-Epoxy-Coal-Tar-Pitch_62538348089.html.
3. *Concrete Prices*. Available at: <https://www.sci99.com/monitor-94896214-0.html>; <https://www.statista.co%0Am/statistics/219339/us-prices-of-cement/%0A>.
4. Zhou, L. et al. 3D self-assembly of aluminium nanoparticles for plasmon-enhanced solar desalination. *Nat. Photonics* 10, 393–398 (2016).
5. Singh, S. C. et al. Solar-trackable super-wicking black metal panel for photothermal water sanitation. *Nat. Sustain.* 3, 938–946 (2020).
6. Ni, G. et al. A salt-rejecting floating solar still for low-cost desalination. *Energy Environ. Sci.* 11, 1510–1519 (2018).
7. Wu, L. et al. Highly efficient three-dimensional solar evaporator for high salinity desalination by localized crystallization. *Nat. Commun.* 11, 1–12 (2020).
8. Morciano, M., Fasano, M., Boriskina, S. V., Chiavazzo, E. & Asinari, P. Solar passive distiller with high productivity and Marangoni effect-driven salt rejection. *Energy Environ. Sci.* 13, 3646–3655 (2020).
9. Xia, Y. et al. Spatially isolating salt crystallisation from water evaporation for continuous solar steam generation and salt harvesting. *Energy Environ. Sci.* 12, 1840–1847 (2019).

10. Xu, W. *et al.* Flexible and Salt Resistant Janus Absorbers by Electrospinning for Stable and Efficient Solar Desalination. *Adv. Energy Mater.* 8, (2018).
11. Kuang, Y. *et al.* A High-Performance Self-Regenerating Solar Evaporator for Continuous Water Desalination. *Adv. Mater.* 31, (2019).
12. He, S. *et al.* Nature-inspired salt resistant bimodal porous solar evaporator for efficient and stable water desalination. *Energy Environ. Sci.* 12, 1558–1567 (2019).
13. Chen, X. *et al.* Sustainable off-grid desalination of hypersaline waters using Janus wood evaporators. *Energy Environ. Sci.* 14, 5347–5357 (2021).
14. Xu, N. *et al.* A water lily-inspired hierarchical design for stable and efficient solar evaporation of high-salinity brine. *Sci. Adv.* 5, eaaw7013 (2019).
15. Zhang, Y. *et al.* Manipulating unidirectional fluid transportation to drive sustainable solar water extraction and brine-drenching induced energy generation. *Energy Environ. Sci.* 13, 4891–4902 (2020).

References for Response to Reviewer 1

1. Mekonnen, M. M. & Hoekstra, A. Y. Four billion people facing severe water scarcity. *Sci. Adv.* 2, e1500323 (2016).
2. Tao, P. *et al.* Solar-driven interfacial evaporation. *Nat. Energy* 3, 1031–1041 (2018).
3. Wang, Z. *et al.* Pathways and challenges for efficient solar-thermal desalination. *Sci. Adv.* 5, eaax0763 (2019).
4. Zhang, L. *et al.* Passive, high-efficiency thermally-localized solar desalination. *Energy Environ. Sci.* 14, 1771–1793 (2021).
5. Zhang, Y., Xiong, T., Nandakumar, D. K. & Tan, S. C. Structure Architecting for Salt-Rejecting Solar Interfacial Desalination to Achieve High-Performance Evaporation With In Situ Energy Generation. *Adv. Sci.* 7, 1903478 (2020).
6. Menon, A. K., Haechler, I., Kaur, S., Lubner, S. & Prasher, R. S. Enhanced solar evaporation using a photo-thermal umbrella for wastewater management. *Nat. Sustain.* 3, 144–151 (2020).
7. Cooper, T. A. *et al.* Contactless steam generation and superheating under one sun illumination. *Nat. Commun.* 9, 5086 (2018).

Reviewer 2

Natural and passive convection is probably the most widely applied approach to realize practical implementation of salt-rejecting solar desalination, but previous studies rarely provide useful in-depth exploration to dissect this process, thus I find encouraging findings regarding this point in the current manuscript by Zhang et.al. Unlike most of previously reported solar evaporators, the developed solar absorber is completely immersed below water level, but still capable of realizing a very high conversion efficiency (>90%), which is an interesting and surprising finding and worthy of such a study. This novel and effective salt rejection strategy presented in this work is also of significance to facilitate understanding of the topic of sustainable solar desalination. I thus think it is highly suitable for publication in Nature Communications, but some key issues should be carefully addressed, such that its quality could get further improved, specifically as follows:

Response: We are glad that the Reviewer thinks our work is highly suitable for publication. We really appreciate the careful review and insightful comments of the Reviewer, which has helped us improve the quality of this work. We carefully addressed all of the Reviewer's comments and provide detailed response as shown below.

1. The developed solar-evaporation configuration works in a way quite different from traditional solar basin (Adv. Sci. 2020, 7, 1903478.), as the black liner of the current device is much closer to water level, and of course, with proper insulation beneath it. Thus, the total volume of salt water within this confined space is important to suppress volumetric heating and following conduction loss, and at the same time, enable sufficient mass exchange, i.e., the convection, to avoid salt formation. Then the thickness of the confined water layer is considered a key parameter to this trade-off, but this critical investigation was not provided here. How was it determined in the current manuscript? and by changing it, how will it influence the performance? Although the conclusion is obvious that an optimal thickness is attainable under a given configuration, these questions should be carefully discussed.

Response: We first thank the Reviewer for providing this useful review article (Adv. Sci. 2020, 7, 1903478), which provides a comprehensive summary for the recent solar evaporation designs. We have cited this important paper as Ref. 7 in the original main text to support our work. We further cited this review paper to support that wick structures are commonly used for thermally localized solar evaporation in Page 3, Line 13 of the revised main text,

“In a typical thermally localized solar evaporation device (Fig. 1a), a capillary wick structure is used to enable the solar-thermal conversion, thermal localization, and passive water supply^{3,5,7,8,20}.”

We highly agree with the Review that it is necessary to provide design guidelines for the confined water layer thickness. Although the macrochannel diameter is the most important design parameter to enable the simultaneous thermal localization and salt rejection, as expected by the Reviewer, confined water layer thickness also affects salt transport and the 5 mm thickness used in our design is based on a careful optimization.

In the revised main text, we discussed the role of confined water layer thickness in salt transport and solar evaporation to justify the 5 mm thick confined water layer used in our design. With mechanistic modeling, we show that although the confined water layer thickness has insignificant impact on the average salt concentration of the entire confined water layer, it strongly affects the uniformity of salt spatial distribution. Uniform salt concentration is highly important since salt crystallization always occurs at the position with the highest concentration. In general, we show that a very thin confined water layer leads to highly nonuniform salt concentration with low concentration around the macrochannel and high concentration close to the external insulation ring (Fig. R4a). Since the salt transport resistance along the confined water layer $R_{s,\parallel}$ (in-plane direction) is inversely proportional to the confined water layer thickness t (i.e., $R_{s,\parallel} \sim 1/t$), reducing t increases $R_{s,\parallel}$ and hence leads to the highly nonuniform salt concentration (Fig. R4a). On the other hand, although increasing t is desirable for the in-plane salt transport, too thick confined water layer will increase the cross-plane transport resistance $R_{s,\perp}$ (i.e., $R_{s,\perp} \sim t$) and thus enlarge nonuniformity of salt concentration crossing the confined water layer. The large thickness of confined water layer also increases heat loss due to the increase of sidewall area. The 5 mm thick confined water layer used in our design enables the highest uniformity of salt concentration (Fig. R4b) and avoids significant heat loss.

Fig. R4. Salt concentration profiles in (a) 1 mm and (b) 5 mm thick confined water layers obtained from simulations. Macrochannel diameters for both cases are 2.5 mm and solar illumination is one sun. The salt concentration profiles were taken at ≈ 0.5 hours after the simulation started.

To better elucidate the effect of confined water layer thickness, we added the following discussion in Page 11, Line 1 of the revised main text,

“Although the diameter of macrochannels is the most important design parameter to achieve the simultaneous thermal localization and salt rejection, careful optimization of the confined water layer thickness, macrochannel arrangement, and thermal insulation thickness was also performed to provide complete design guidelines for the confined water layer structure. In particular, the confined water layer thickness strongly affects the uniformity of salt concentration. Nonuniform salt concentration is undesirable for practical applications because salt crystallization will always occur at the position with the highest concentration. With mechanistic modeling of evaporating 20

wt% brine under one sun illumination, we showed that very thin confined water layer (< 3 mm) will lead to significant spatial nonuniformity of salt concentration (≈ 1 wt% salinity difference) due to insufficient in-plane salt transport. Meanwhile, too thick of a confined water layer (> 10 mm) will also increase the nonuniformity because of the reduced cross-plane salt transport (see Supplementary Figure 8a for details). In addition, the thick confined water layer will increase heat loss due to the increase of sidewall area (see Supplementary Figure 8b for details). For these reasons, a 5 mm thick confined water layer was used in our design to ensure an optimal uniformity of salt concentration and insignificant heat loss.”

To further support the discussion in the main text, we show detailed simulations and analysis in Supplementary Figure 8 of the revised Supplementary Information (Page 13),

Supplementary Figure 8. Effects of the confined water layer thickness on (a) salt transport and (b) heat loss predicted by simulations. Solar evaporation of 20 wt% brine under one sun illumination was simulated. The macrochannel diameter is 2.5 mm. (a) Nonuniformity of salt concentration $\Delta\phi_{\max}$ as a function of the confined water layer thickness at the transition point to the quasi-steady state (i.e., ≈ 0.5 hours after the simulation started). $\Delta\phi_{\max}$ is the maximum value of salt concentration difference in the confined water layer, which is defined as the difference between the highest salt concentration and the lowest salt concentration. Insets: top view of salt concentration profiles with 1 mm and 5 mm thick confined water layers. Nonuniform salt concentration is undesirable because salt crystallization will first occur at the position with the highest salt concentration. $\Delta\phi_{\max}$ first rapidly decreases and then slightly increases with the increase of the confined water layer thickness, leading to a minimum point approximately at 5 mm, which was selected for our design. The large nonuniformity of salt concentration occurs at a low confined water layer thickness (e.g., 1 mm), because the resistance of salt transport along the confined water layer (i.e., the lateral transport) is inversely proportional to its thickness, resulting in a low salt concentration near the macrochannel with a high salt concentration close to the external floating ring (see the inset of (a)). The slight increase in salt concentration nonuniformity at the large thickness (e.g., 10 mm) is attributed to the increased resistance of salt transport crossing the confined water layer (i.e., the vertical transport). (b) Heat loss through the sidewall of a floating ring as a function of the confined water thickness. Heat loss increases linearly due to the increase of sidewall area.

In addition to the confined water layer thickness, in the revised manuscript, we also show (1) experimental characterization of the impact of macrochannel diameter, (2) effect of macrochannel spacing on the salt rejection, and (3) solar evaporation performance of the nearly saturated brine (25 wt%). We hope these detailed experiments and simulations will provide a clearer design guideline for the confined water layer structure. For the convenience of the Reviewer, we summarize the key findings and changes as follows.

(1) In our original manuscript, we demonstrate that the macrochannel diameter is the most important design parameter to enable the simultaneous thermal localization and salt rejection, where too small diameter cannot initiate the natural convection while too large diameter will lead to significant heat loss. We obtained the optimal macrochannel diameter (2.5 mm) with mechanistic modeling and experimentally demonstrated the resulting superior solar evaporation performance. In the revised manuscript, we further experimentally demonstrated the impact of macrochannel diameter. Specifically, we performed solar evaporation experiments of 20 wt% brine using confined water layer structures with 1 mm and 5 mm macrochannel diameters, where we observed salt crystallization on the 1 mm diameter design and reduced thermal localization on the 5 mm diameter design. Both designs show lower solar-to-vapor conversion efficiency than the optimal design with 2.5 mm macrochannel diameter. This experimental characterization further strengthens the design strategy elucidated in our original manuscript. In Page 10, Line 22 of the revised main text, we added the following discussion,

“We experimentally validated this regime by evaporating 20 wt% brine on three confined water layer structures with 1 mm, 2.5 mm, and 5 mm diameter macrochannels (see Supplementary Figure 6, 7 for details). Significant salt crystallization and increased heat loss through macrochannels were observed on the 1 mm and 5 mm diameter macrochannel designs, respectively, both of which reduce the solar-to-vapor conversion efficiency (see Supplementary Figure 7 for details). Guided by the modeling and experimental results, we chose $d = 2.5$ mm as the macrochannel diameter for our prototype.”

The corresponding experimental characterizations are shown in Supplementary Figure 6 and 7 in the Page 12 and 13 of the revised Supplementary Information,

Supplementary Figure 6. Top view of the (a) floating ring, insulation foams with (b) 1 mm, (c) 2.5 mm, and (d) 5 mm macrochannel diameters used for experimental characterizations. Unit is in mm.

Supplementary Figure 7. Solar evaporation of 20 wt% brine under one sun illumination. Three confined water layer structures with 1 mm, 2.5 mm, and 5 mm diameter macrochannels were tested (see Supplementary Figure 6 for detailed dimensions). (a) Temperature response of the confined water layer with different macrochannel diameters. The steady state temperature decreases with macrochannel diameters due to the increased heat loss through the floating insulation, which validates our theoretical prediction in Fig. 3a of the main text. (b) Corresponding mass change of the evaporation setup as a function of time. Confined water layer structure with 2.5 mm diameter macrochannels shows the highest evaporation rate, which was chosen for our final design. The reduced evaporation rate of the 1 mm diameter macrochannel design is attributed to the salt crystallization and increased convective heat loss due to the elevated water layer temperature, whereas the reduced evaporation rate of the 5 mm diameter macrochannel design is induced by the increased conductive heat loss through macrochannels. Time-lapse images of the confined water layer structure with (c) 1 mm and (d) 5 mm diameter macrochannels during a 3-hour continuous solar evaporation. Salt crystallization was observed on the 1 mm diameter macrochannel design after the second hour, because natural convection enhanced salt rejection cannot be initiated with small macrochannel diameters. Scalebar: 2 cm.

(2) In our original manuscript, we chose macrochannel arrangement with five macrochannels and 9 mm spacing (see Supplementary Figure 6c) to ensure a uniform salt concentration in the confined

water layer. In the revised manuscript, we provide more justifications for the macrochannel arrangement. With mechanistic modeling, we investigated the nonuniformity of salt concentration as a function of macrochannel spacing. In general, macrochannel spacing weakly affects the salt transport whereas there is still an optimal spacing close to 9 mm for the highest salt concentration uniformity, which was used for our design. In Page 11, Line 16 of the revised main text, we added the following discussion,

“Five macrochannels were fabricated where one of them was located at the center of the thermal insulation and the other four were 9 mm away from the center (see Supplementary Figure 2). We chose this configuration to further improve the uniform salt concentration in the confined water layer. With mechanistic modeling, we show that although macrochannel spacing plays a relatively insignificant role in salt transport, the 9 mm macrochannel spacing is desirable to enable the highest uniformity of salt concentration (see Supplementary Figure 9 for details).”

The corresponding detailed results are shown in Supplementary Figure 9 in Page 15 of the revised Supplementary Information,

(3) Supplementary Figure 9. Effects of macrochannel spacing on the nonuniformity of salt concentration predicted by simulations. Solar evaporation of 20 wt% brine under one sun illumination was simulated. The macrochannel diameter is 2.5 mm and the confined water layer thickness is 5 mm. (a) Top view of three representative insulation foams with 4 mm, 9 mm, and 13 mm macrochannel spacing used for simulations. Unit is in mm. (b) Nonuniformity of salt concentration as a function of macrochannel spacing. In general, $\Delta\phi_{\max}$ weakly depends on the macrochannel spacing, but too large of a macrochannel spacing will lead to an increase of salt concentration nonuniformity. 9 mm spacing was chosen for our design to achieve the optimal macrochannel arrangement.

(4) In our original, we demonstrate high-efficiency solar evaporation of 20 wt% brine without salt crystallization using the confined water layer structure. It is interesting to see the limit of our design by pursuing even higher salinity. For this reason, in the revised manuscript, we performed solar evaporation of 25 wt% brine, which is close to the saturation point (26 wt%) at room temperature. During 6-hour continuous operation under one sun illumination, no salt crystallization was observed, and the solar-to-vapor conversion efficiency was 67%. This experiment of nearly saturated brine further demonstrates the superior salt rejection capability of the confined water layer structure. In Page 19, Line 10 of the revised main text, we added the following discussion,

“Moreover, to show the limit of the confined water layer structure, we performed 6-hour continuous solar evaporation of 25 wt% brine, which is approaching the saturation level of NaCl in water at room temperature (≈ 26 wt%). No salt crystallization was observed during the 6-hour test and 67% solar-to-vapor conversion efficiency was demonstrated (see Supplementary Figure 12 for details).”

The corresponding experimental characterizations are shown in Supplementary Figure 12 in Page 18 of the revised Supplementary Information,

Supplementary Figure 12. 6-hour continuous solar evaporation of 25 wt% brine under one sun illumination. (a) Temperature response of the confined water layer. (b) Corresponding mass change of the evaporation setup as a function of time. (c) Time-lapse images of the confined water layer structure during the 6-hour continuous solar evaporation. Scalebar: 2 cm. The brine concentration (25 wt%) is approaching to the saturation point (26 wt%) at room temperature. No salt crystallization was observed during the 6-hour continuous solar evaporation, demonstrating the superior salt rejecting performance of the confined water layer structure. The corresponding solar-to-vapor conversion efficiency is 67%.

2. The solar absorbance is 0.953 as reported by the authors, I suppose it was measured in completely dry state, since in the actual scenario, a confined water layer is covering the solar absorber, and with this reflective water film, the overall absorbance could not be so high. In the simulations, 0.953 was taken into account as well, if not considering this likely reduction in solar absorption caused by the water film, the calculated outcome might not be accurate any longer.

Response: Thanks for this comment. We agree with the Reviewer that the presence of the water layer could affect the solar absorption in practical operations. However, due to the low refractive index mismatch between water ($n_{water} = 1.33$) and air ($n_{air} = 1$), water is a weak scatterer for visible light, and the reduction of visible light absorption due to the reflection of water layer is not significant. Specifically, the total reflection of visible light due to the water layer can be estimated by Fresnel's law,

$$R = \left| \frac{n_{water} - n_{air}}{n_{water} + n_{air}} \right|^2 = \left| \frac{1.33 - 1}{1.33 + 1} \right|^2 \approx 2\% \quad (R1)$$

where R is the reflectivity of water layer. Note that 2% of reflection is a conservative estimation (*i.e.*, the upper bound) because in practical operations, there is saturated vapor mixed with air above the water layer, which has a better refractive index match with water than the dry air used for our calculation in Eq. R1. Since the visible light accounts for approximately 43% of the total energy of solar radiation (Fig. R5a) [1], solar energy reflected by the water layer is less than 1% of the total solar radiation.

On the other hand, water is highly absorbing in the entire infrared (IR) regime (> 1000 nm wavelength as shown in Fig. R5b). Therefore, the presence of water will further enhance the absorption in the IR band of solar illumination, which accounts for 52% of the total solar energy (Fig. R5a) [1]. In addition, the water layer on the top of the solar absorber can further absorb the thermal radiation from the solar absorber. Although it is challenging to precisely quantify the role of the water layer in the total solar absorption, considering all of the above mechanisms including both the weak reflection to visible light and strong absorption to IR light, we believe 95.3% solar absorbance can be a reasonable estimation for our confined water layer design.

Fig. R5. (a) Solar spectrum and energy distribution in ultraviolet (UV), visible (VIS), and infrared (IR) regimes [1]. (b) Water absorption as a function of wavelength [2].

To make this point clear, we added the following explanations in Page 25, Line 9 of the revised main text,

“Note that the solar absorptance was characterized in a dry state without water layer on the top of the solar absorber. The presence of the water layer will lead to a weak reflection to visible light ($\leq 2\%$ of the visible light according to Fresnel’s law) while enhanced absorption to IR light (> 1000 nm wavelength). Considering this combined effect, 95.3% can be a reasonable estimation for actual solar absorption during practical operations, which was used in our design and analysis.”

3. With this overestimated solar absorbance, which requires further confirmation by the authors, the overall solar efficiency was 91% as reported by the authors, which is surprisingly high, especially considering that the solar absorber was immersed beneath the water level instead of at water/air interface. Thus, heat losses due to different outlets such as conduction, radiation, etc. that could account for this 91% must be specified and analyzed.

Response: We agree with the Reviewer that it is challenging to achieve above 90% solar-to-vapor conversion efficiency. We achieved the high solar-to-vapor conversion efficiency by carefully optimizing the overall heat and mass transport. To address the Reviewer’s concern, we provide an energy balance analysis in this Response. As mentioned by the Reviewer, heat loss is induced by conduction, convection, and radiation. The thermal conductivity of the floating insulation used in our design is approximately $k_s = 0.03$ W/mK. The steady state temperatures of the confined water layer, bulk water, and ambient environment are approximately 37 °C, 26 °C, and 25 °C, respectively, which are obtained from our experiment in Fig. 5c of the main text. The conductive loss through the floating thermal insulation is thus estimated by Fourier’s law,

$$q''_{cond} = k_s \frac{\Delta T_w}{t_s} \approx 0.03 \times \frac{11}{0.025} = 13.2 \text{ W/m}^2 \quad (\text{R2})$$

where ΔT_w is the temperature difference between the confined water layer and the bottom bulk water. $t_s = 25$ mm is the thickness of the floating insulation used in this work. The convective heat loss is mainly induced by natural convection, which can be estimated by the classical heat transfer correlations. Specifically, the Rayleigh number Ra of the natural convection on the top of the confined water layer is given by,

$$Ra = GrPr = \frac{g\beta(T_w - T_\infty)D^3}{\nu^2} Pr = \frac{9.81 \times \frac{1}{304.15} \times 12 \times 0.031^3}{(16.5 \times 10^{-6})^2} 0.7 = 2.96 \times 10^4 \quad (\text{R3})$$

where Gr and Pr are the Grashof number and the Prandtl number, respectively. g , β , T_w , T_∞ , D , and ν are gravitational acceleration, coefficient of volume expansion, confined water layer

temperature, ambient temperature, evaporator diameter, and kinematic viscosity of air, respectively. The Nusselt number Nu for the natural convection can be estimated by the following widely used correlation [3],

$$Nu = 0.54Ra^{\frac{1}{4}} = 7. \quad (\text{R4})$$

Therefore, the heat transfer coefficient h_{conv} for the convective heat loss is,

$$h_{conv} = \frac{Nuk_a}{D} = 5.8 \text{ W/m}^2\text{K}, \quad (\text{R5})$$

and the resulting convective heat loss is given by,

$$q''_{conv} = h_{conv}(T_w - T_{\infty}) = 70 \text{ W/m}^2. \quad (\text{R6})$$

The radiative heat loss from the confined water layer is negligible because the saturated vapor above the confined water layer is a participating media, which induces a strong greenhouse effect and suppresses the thermal radiation [4]. As a result, the total heat loss accounts for approximately 8.3% of the total input solar flux (*i.e.*, 1000 W/m^2). In general, this energy balance analysis conserves the total energy well. If the total solar absorption is 95.3%, the resulting solar-to-vapor conversion efficiency will be approximately 87%, which shows reasonably good agreement with the experimentally measured efficiency (91%) from the mass loss. The discrepancy between the energy balance estimation and experimental measurement can be attributed for the following three reasons.

(1) The convective heat loss may be overestimated. The empirical correlation for natural convection (Eq. R4) describes the heat transfer from a perfectly flat surface, where most of heat transfer occurs at the edge region with the lowest thermal boundary layer thickness (Fig. R6a). However, in our design, the confined water layer structure is bounded by the external insulation ring, which could elevate the thermal boundary layer at the edge and hence reduce the convective heat loss (Fig. R6b).

Fig. R6. Schematic of the natural convection occurring on (a) the perfectly flat surface and (b) the confined water layer structure. Thermal boundary layer at the edge region could be elevated by the external insulation ring, which leads to lower convective heat loss.

(2) The total solar absorption may be underestimated. As discussed above, although the reflection of visible light due to the confined water layer can result in up to 1% reduction of the total solar absorption, the enhanced absorption to the IR light might compensate for the energy loss in the visible regime and further increase the total solar absorption. For example, the IR absorption of our solar absorber in the dry state is $\approx 95\%$. If the IR absorption approaches to 100% owing to the presence of water layer, the resulting increase of total energy absorption will be $\approx 2.5\%$.

(3) There may be additional solar absorption outside the confined water layer. To avoid additional solar absorption, we used weakly absorbing materials for our design, including the white polystyrene foam for the external insulating ring and the transparent acrylic for the container of bulk brine (Fig. R7). We also used highly solar reflective aluminum foil to cover all of the rest of the surfaces of our experimental setup (Fig. R7) and carefully reduced the solar beam size close to the absorber size with an aperture. However, these materials are not ideal and can still absorb a small amount of solar energy. This additional inevitable solar absorption, occurring in both the well-controlled experiments and practical implementation, could be another reason for the small discrepancy between the energy balance analysis and experimental measurement.

Fig. R7. A representative image of the confined water layer structure, where except for the solar absorber, weakly absorbing and highly reflective materials were used for the design of experimental setup to avoid additional solar heating.

4. Clarifications are needed regarding the demonstrations shown in Fig. 3C and 3D: 1) this dripping action for the two tests should be identically synchronous for a strict comparison, but there was still tracer liquid left in the syringe at 1s (3C) while the injection in 3D was completed already; 2) It is not clear what the inset caption (DI water and Brine) refers to (the bulk water or the dyed water?); 3) it is hard to tell how convection effect makes a difference as the red fluid reached to the bottom slowly in 3C, while the other one (3D) was obviously faster but gathered, besides, the shade of dyed water (one is light, one is dark) is also a problem to make a direct and clear comparison.

Response: Thanks for these comments. We first apologize for the wrong order of Fig. 3c and d in our original main text, which is inconsistent with the corresponding discussion and captions. In fact, the original Fig. 3d represents the dripping of 20 wt% brine, which was discussed before Fig. 3c. In the revised main text, we corrected the order of Fig. 3c and 3d and updated Fig. 3 in Page

13 of the revised main text. For the convenience of the Reviewer, we copied the revised Fig. 3 as Fig. R8 of this Response. Here we provide a point-to-point response to the above three comments of the Reviewer.

(1) We apologize for the inaccurate timeline due to the poorly aligned video frames. The zeroth second in Fig. 3c and 3d was defined as the moment when the dye colored liquid is completely infused onto the confined water layer. However, since each second consists of 30 video frames, the frame that was picked for the zeroth second was misaligned a little bit. We corrected this problem by carefully realigning all of the frames and updated Fig. 3 in the revised main text (Fig. R8 in this Response).

Fig. R8. A copy of revised Fig. 3 in the main text. The frames of c and d were carefully realigned. (a) Salinity and temperature difference across the macrochannel as a function of the macrochannel diameter. (b) Confined water layer salinity as a function of time after dripping 2.3 mL 20 wt%

brine in an isothermal condition. (c) Time-lapse image of the transport of 20 wt% brine visualized by food dye. (d) Time-lapse image of the transport of DI water visualized by food dye.

We would like to clarify that the dye colored liquid has been completely injected onto the confined water layer at the zeroth second for both Fig. R8c and d, because the plungers have already moved to the end of the barrel for both two syringes. The remaining liquid containing in the hubs of two syringes cannot be injected. Although the syringe can still be seen in the first second of Fig. R8d, no additional dye colored liquid was injected. Since we operated these two syringes manually, it is challenging to move the syringe out of the field of view synchronously. To avoid potential confusion, we provided a video for the dye colored liquid injection visualization in the revised manuscript as a Supplementary Video. It can be seen from the video that the liquid injection was completed at the sixth second of the video, which was chosen as the zeroth second of the liquid transport visualization shown in Fig. R8c and d. After the sixth second of the video, no additional liquid was injected onto the confined water layer even though the syringe for Fig. R8d was not moved out of the field of view in time. For the convenience of the Reviewer, we created a snapshot for the sixth second of the video into Fig. R9.

Fig. R9. A snapshot for the sixth second of the dripping test video, which shows that the dripping has already been completed for both the DI water and saline water.

We also added a brief description for the Supplementary Video, which can be seen in the Description of Additional Supplementary Files,

“Video 1. Dripping tests of 1 mL food dye colored deionized (DI) water (left) and 20 wt% brine onto the confined water layer. Water reservoir initially contained DI water only. The dripping process was completed at the sixth second. The zeroth second for the liquid transport visualization shown in Fig. 3c,d of main text corresponds to the sixth second of this video.”

In addition, we would like to mention that the dye colored liquid tests shown in Fig. R8c and d are mainly for qualitative visualization rather than quantitative characterization. In fact, we provided more rigorous and quantitative characterization in Fig. R8b, where we directly measured the salinity variation as a function of time after the dripping. The good agreement between the experiment and simulation validated the strong natural convection effect.

(2) We apologize for the confusion due to our unclear labeling. In our dripping tests, we dripped DI water and 20 wt% brine onto the confined water layer, where the reservoir initially contained DI water only. In fact, we explained this experimental condition in Page 11, Line 24 of our original manuscript,

“The experiment was in an isothermal condition to decouple the thermal effect. The reservoir initially contained deionized (DI) water only. We uniformly dripped 2.3 mL 20 wt% brine onto the confined water layer within 15 s (inset of Fig. 3b)...”

To avoid confusion, we removed the improper labels in Fig. 3c and d of the main text, which can be seen in Fig. R8c and d in this Response. Alternatively, we explained the experimental condition in the caption of Fig. 3 (Page 13, Line 11) and the description of Supplementary Video,

“c Time-lapse image of the transport of 20 wt% brine visualized by food dye. d Time-lapse image of the transport of DI water visualized by food dye. For both visualization tests, the reservoir initially contained DI water only.”

(3) We presume the confusion of the Reviewer for this point is due to the wrong order of Fig. 3c and d in our original manuscript. In our revised version, Fig. 3c and d (*i.e.*, Fig. R8c and d, respectively) represent the dripping tests of 20 wt% brine and DI water onto the confined water layer, respectively. The natural convection effect, *i.e.*, the liquid flow *from the confined water layer to the bottom of the reservoir* driven by the density gradient of liquid, is visualized by the dye colored flow. As mentioned by the Reviewer, the dye colored 20 wt% brine flows to the bottom much faster, indicating a stronger natural convection *from the confined water layer to the bottom of the reservoir*. The gathering of the dye colored liquid at the bottom is due to the higher liquid density of 20 wt% brine, which directly proves that the colored saline water has already been carried to the bottom by the convective flow. Therefore, we believe the dripping of dye colored brine can visualize the natural convection effect.

The different colors shown in Fig. R8, *i.e.*, dark red in Fig. R8c while light red in Fig. R8d, are also originated from the significantly different strength of natural convection. In fact, as shown in the Supplementary Video and Fig. R10 (*i.e.*, the first frame of the video), we used the same concentration of dye to mark the DI water and saline water in the syringes. However, since the natural convection effect of the DI water is too weak, most of the liquid containing the high concentration dye remained in the confined water layer throughout the test, leading to the dilute tracks shown in Fig. R8d. To make this point clear, we added the following discussion in Page 12, Line 11 of the revised main text,

“In addition, despite the same concentration of red food dye used for both the 20 wt% brine and DI water dripping, most of the food dye colored DI water remained in the confined water layer throughout the entire test because of the insignificant convective flow, leading to the much lighter colored liquid trajectories in Fig. 3d than that in Fig. 3c (Supplementary Video).”

Fig. R10. The first frame of the dripping test video, which shows the same concentration of dye in syringes used to mark the DI water and saline water.

5. A particular concern regarding the immersed solar absorber is the bio-fouling issue. It is not like other capillary driven systems where bio-impurities could be rejected simply by spontaneous capillary filtration, such as the unidirectional transportation (as reported in ACS Energy Lett. 2020, 5, 11, 3397–3404; Desalination. 2021, 515, 115192; ACS Appl. Mater. Interfaces 2021, 13, 32, 38405–38415), how can the current device prevent itself from bio-fouling?

Response: We agree with the Reviewer that the bio-fouling could be another critical issue for the reliable solar evaporation devices and appreciate the Reviewer for suggesting these three important references. We believe that engineering the convective flow, including both the unidirectional flow and the natural convection, can further enhance the anti-biofouling performance, because the biocontainment can be carried by the convective flow and rejected back to the bulk liquid. However, since anti-biofouling performance is not the main purpose of the current study, we did not perform in-depth experimental characterizations. To highlight the importance of biofouling issue, we cited the suggested references as Refs. 42-44 and added the following sentence in Page 24, Line 6 of the revised main text. We hope it will inspire more relevant works along this direction.

“In addition, it is possible to improve the resistance to biofouling by taking advantage of the convective flow⁴²⁻⁴⁴, which requires further investigation in future works.

42. Zhang, Y. et al. *Guaranteeing Complete Salt Rejection by Channeling Saline Water through Fluidic Photothermal Structure toward Synergistic Zero Energy Clean Water Production and In Situ Energy Generation*. *ACS Energy Lett.* 5, 3397–3404 (2020).

43. Li, Y. et al. *Composite hydrogel-based photothermal self-pumping system with salt and bacteria resistance for super-efficient solar-powered water evaporation*. *Desalination* 515, 115192 (2021).

44. Peng, H., Wang, D. & Fu, S. *Unidirectionally Driving Nanofluidic Transportation via an Asymmetric Textile Pump for Simultaneous Salt-Resistant Solar Desalination and Drenching-Induced Power Generation*. *ACS Appl. Mater. Interfaces* 13, 38405–38415 (2021).”

6. Others: 1) information like the model numbers and brands of instruments employed in the work are found twice in the main text and in the experimental section, which seems of no need, it would be better to just leave them in the experimental section; 2) same symbol was used for salinity and diameter (Fig.S2); 3) the unit of salinity is not consistent (wt% in the main text, w% in SI).

Response: We really appreciate for pointing out these critical issues. Here we provide our point-to-point response to the above three comments.

(1) As suggested by the Reviewer, we moved all of the brands and model numbers of instrument from the main text to the Method section in the revised main text.

(2) Thanks for pointing this out. We removed the diameter symbol “ ϕ ” from Supplementary Figure 2. For the convenience of the Reviewer, we copied the revised Supplementary Figure 2 as below.

Supplementary Figure 2. *Top view of the floating ring, insulation foam and copper sheets used for the waterjet machining. The dimensions were determined by the optimization process. Unit is in mm.*

(3) Thanks for pointing this out. We corrected the unit of salinity to “wt%” for all figures. For the convenience of the Reviewer, we copied the revised Supplementary Figure 1 as below.

Supplementary Figure 1. Optimization of the macrochannel size and the insulation layer thickness to enable the simultaneous thermal localization and salt rejection. A single-channel unit with axial symmetry was simulated for the initial design optimization. Temperature and salinity profiles in the second hour are shown with different macrochannel sizes (1 mm to 5 mm) and insulation thicknesses (6.4 mm to 38.1 mm)

References for Response to Reviewer 2

1. Kruse, O. *et al.* Photosynthesis: a blueprint for solar energy capture and biohydrogen production technologies. *Photochem. Photobiol. Sci.* 4, 957–969 (2005).
2. Bec, K. and Huck C. Breakthrough Potential in Near-Infrared Spectroscopy: Spectra Simulation. A Review of Recent Developments. *Front. Chem.* 7, 48 (2019).
3. Mills, A. F. Heat Transfer, Prentice Hall (1999).
4. Zhao, L. *et al.* Harnessing Heat Beyond 200 °C from Unconcentrated Sunlight with Nonevacuated Transparent Aerogels. *ACS Nano* 13, 7508–7516 (2019).

Reviewer 3

In this article, the authors reported a solar evaporation system based on the employment of wick-free water transportation pathway and a confined water layer. The key point and advance are the realization of thermal localization and salt rejection convection from the view point of fluidic flow optimization. The article is very well written and organized. However, I still have some reservations, on addressing which the article can be improved further.

Response: We really appreciate the Reviewer for the careful review and positive feedback. We believe the Reviewer's comments are highly valuable, which helped us improve the quality of our work. We carefully addressed all of the concerns raised by the Reviewer and provide detailed response as shown below.

1. The authors emphasized the use of the wick-free confined water layer, as salt accumulation phenomenon, especially when evaporating high salinity brine, cannot be fully addressed by the wick-based system. However, it is reasonable to think of a wick-based system with similar arrayed channel configuration and fluidic regulation (Adv. Mater. 2019, 31, 1900498, Reference 16 in this manuscript) where similar or even better performance can be achieved (1.46 kg m⁻² h⁻¹ evaporation rate for DI water, 100 hours continuous salt accumulation free, and 60 days stability, etc.), comparing to this system. The advantage of this wick-free system over the wick-based system seemed not to be clear enough with current data and analyses. Further investigation and optimization are needed to satisfy the high standard of Nature Communications.

Response: Thanks for this comment. As mentioned by the Reviewer, the performance of self-regenerating solar evaporator reported by Kuang *et al.* in Prof. Liangbing Hu's group is highly impressive [1]. However, we believe the wick-free confined water layer structure developed in this work is also highly valuable in terms of the following *three aspects*.

(1) At a fundamental level, to the best of our knowledge, our work is one of the first studies which provides a *fully quantitative understanding of salt transport* during solar evaporation. Although various salt rejecting solar evaporators have been developed in recent years, there lacks a fundamental understanding salt transport, which impedes the predictive and quantitative design to push the fundamental limit. In particular, as we mentioned in the main text, two fundamental problems have not been addressed: (1) how to break the conventional diffusion-limited salt rejection by engineering passive convective flow, and (2) how to understand the interplay of thermal localization and salt rejection due to water confinement and convective flow.

With a mechanistic understanding of salt transport coupled with heat transfer and fluidic flow, for the first time, we demonstrate the opportunity space of *manipulating passive natural convection* to enable simultaneous thermal localization and salt rejection, which was not observed in the previous wick-based solar evaporators. Therefore, we believe our work provides a fresh perspective *at the intersection of solar evaporation and fluidic engineering*, which complements the existing wick-based systems.

In addition, we believe insights gained from our study are also helpful to understand the enhanced salt rejection capability demonstrated in Kuang *et al.*'s self-regenerating solar evaporator. For example, although the natural convection effect was not mentioned in Kuang *et al.*'s work, we believe that the proper millimeter-size channels created on the natural wood evaporator could trigger the natural convective flow, which explains the significantly enhanced salt rejection capability in a few specific morphology designs in Kuang *et al.*'s work (e.g., 1.5 mm diameter channels v.s. 1 mm diameter channels) [1]. This example demonstrates that the fundamental understanding to the coupling of salt transport and fluidic flow is highly valuable to explain and guide relevant designs.

We explained the importance of fundamental salt transport in the Introduction (Page 4, Line 11) and Conclusion sections (Page 23, Line 22) of the main text,

In Page 4, Line 11 of the main text, we mentioned,

“Specifically, there are two fundamental problems to be addressed: (1) Breaking the conventional diffusion-limited salt rejection by engineering passive convective flow. Since significant enhancement of salt rejection can be achieved by introducing convective flow (e.g., Marangoni flow²² and unidirectional flow²³), a quantitative understanding of how to passively initiate a convective flow becomes important. (2) Understanding the interplay between thermal localization and salt rejection due to water confinement and convective flow. Since convective flow also increases heat loss, a guideline to maximize salt rejection while minimizing heat loss is required²⁰.”

In Page 23, Line 22 of the main text, we mentioned,

“We believe that engineering passive fluidic flow is a promising while not fully explored avenue toward significant enhancement of salt rejection. The fundamental understanding of salt transport plays a central role in manipulating the fluidic flow. This work develops a mechanistic model by coupling the salt transport with the fluidic flow and heat transport to quantitatively guide the evaporator design. We show that the natural convection due to the salinity gradient can be passively triggered by carefully engineering the macrochannels in the thermal insulation. More importantly, owing to the two orders of magnitude difference between the mass diffusivity of salt in water and thermal diffusivity of water, we theoretically and experimentally identified a regime where the convective flow significantly drives salt rejection while inducing negligible additional heat losses – the key to achieve simultaneous thermal localization and salt rejection. We believe this mechanistic model-driven, fully quantitative design approach can serve as general guidelines to interface fluidic flow engineering with various solar evaporation devices.”

(2) For the solar evaporation performance, we agree with the Reviewer that the self-regenerating solar evaporator has very high solar-to-vapor conversion efficiency (1.46 kg/m²/h for 3.6 wt% brine and 75% for 20 wt% brine) and strong salt rejection capability (20 wt% brine without salt crystallization). Although the main purpose of this work is to provide a complementary design pathway with a wick-free structure and natural convection rather than showing the “best” performance, we believe the solar evaporation performance demonstrated in this work is at least

comparable to or even better than the self-regenerating solar evaporator reported by Kuang *et al.* [1]. We elaborate on this with the following three points.

(a) The highest solar evaporation rate and best salt rejection performance were achieved on different solar evaporator designs in Kuang *et al.*'s work [1]. In Kuang *et al.*'s work, the highest solar evaporation rate with 1.46 kg/m²/h was demonstrated on the “Wood/CNT island evaporator” whereas the optimal salt rejection with 75% solar-to-vapor conversion efficiency was demonstrated on the “Carbonized Wood evaporator with 1.5 mm diameter holes”. However, in our work, we show the performance of DI water, 3.5 wt% brine, and 20 wt% brine evaporation on *the same design with the same macrochannel structure*. Therefore, we believe it might be not reasonable to compare the solar evaporation rate and salt rejection that were optimized separately with our design.

(b) For evaporating 20 wt% brine, our work shows a higher solar-to-vapor conversion efficiency (81%) than that of Kuang *et al.*'s work (75%) [1]. One of the key contributions of our work is evaporating concentrated brine with high efficiency. Although both works created channels to enable the evaporation of highly concentrated brine, with a quantitative understanding of salt transport, we are able to further optimize the design parameter to approach to even higher solar-to-vapor efficiency. For a similar salt rejection performance (*i.e.*, 20 wt% brine), the area ratio of macrochannels in our work is only 3%, which is much smaller than that of Kuang *et al.*'s design (7%). For this reason, heat loss through macrochannels can be further reduced, which might be the reason of the improved solar-to-vapor conversion efficiency.

We agree with the Reviewer that the 100-hour continuous solar evaporation is indeed very impressive and important for future solar evaporators. However, since the solar evaporation technology is targeting applications with natural sunlight, which follows the day-night cycle, the continuous applicable daily solar flux typically lasts for less than 6 hours as also shown in our outdoor test. Therefore, we believe enabling the 6-to-8-hour continuous solar evaporation each day is the primary goal of solar evaporator. Our experimental characterizations proved that our design meets this primary goal.

(c) In the revised manuscript, we further demonstrate efficient solar evaporation of nearly saturated brine (25 wt%). To better show the limit of the confined water layer structure, we performed solar evaporation of 25 wt% brine, which is close to the saturation point (26 wt%) at room temperature. During 6-hour continuous operation under one sun illumination, no salt crystallization was observed, and the solar-to-vapor conversion efficiency was 67%. This experiment of nearly saturated brine further demonstrates the superior salt rejection capability of the confined water layer structure. In Page 19, Line 10 of the revised main text, we added the following discussion,

“Moreover, to show the limit of the confined water layer structure, we performed 6-hour continuous solar evaporation of 25 wt% brine, which is approaching the saturation level of NaCl in water at room temperature (\approx 26 wt%). No salt crystallization was observed during the 6-hour test and 67% solar-to-vapor conversion efficiency was demonstrated (see Supplementary Figure 12 for details).”

The corresponding experimental characterizations are shown in Supplementary Figure 12 in Page 18 of the revised Supplementary Information.

Supplementary Figure 12. 6-hour continuous solar evaporation of 25 wt% brine under one sun illumination. (a) Temperature response of the confined water layer. (b) Corresponding mass change of the evaporation setup as a function of time. (c) Time-lapse images of the confined water layer structure during the 6-hour continuous solar evaporation. Scalebar: 2 cm. The brine concentration (25 wt%) is approaching to the saturation point (26 wt%) at room temperature. No salt crystallization was observed during the 6-hour continuous solar evaporation, demonstrating the superior salt rejecting performance of the confined water layer structure. The corresponding solar-to-vapor conversion efficiency is 67%.

(3) At a practical level, our design uses low-cost and widely commercially available materials, which is promising for scalable applications. The natural wood based self-regenerating solar evaporator demonstrated by Kuang *et al.* is highly scalable and attractive [1]. Our work provides an alternative approach to the scalable design, which further decouples the functionalities of solar-thermal conversion, thermal localization, and passive water supply and hence relaxes material constraints. For this reason, the total material cost of a wick-free self-floating confined water layer structure is only $\$2.5\text{-}3.9\text{ m}^{-2}$, which could be lower than the material cost of commercial basswood ($\approx \$50\text{ m}^{-2}$, according to <https://ocoochhardwoods.com/scroll-saw-lumber/basswood/>).

We believe the above three key features make our wick-free design a valuable complement to the existing wick-based solar evaporation approaches.

2. As described in Line 381, Page 17, 91% is the efficiency for DI water. Is the efficiency for 20 wt% NaCl solution also 91%? If not, the statement of “Here, we demonstrate highly efficient (\approx 90% solar-to-vapor conversion efficiency) and salt rejecting (20 weight % salinity) solar evaporation by engineering the fluidic flow in a wick-free confined water layer” in abstract should be revised, otherwise it will lead to misunderstanding.

Response: We apologize for the confusion. The \approx 90% solar-to-vapor conversion efficiency is for DI water evaporation. For different salinities (from 3.5 wt% to 20 wt%), the solar-to-vapor conversion efficiency shown in this work is higher than 80%. Therefore, as suggested by the Reviewer, to avoid misunderstanding, we revised the description to the solar-to-vapor conversion efficiency in the revised Abstract (Page 2, Line 4),

“Here, we demonstrate highly efficient ($>$ 80% solar-to-vapor conversion efficiency) and salt rejecting (20 weight % salinity) solar evaporation by engineering the fluidic flow in a wick-free confined water layer.”

3. The optimization of the macrochannel dimension and the insulation thickness was acquired through simulation only. Experimental results are suggested with simulation results as assist.

Response: We highly agree with the Reviewer that it is necessary to use more experimental results to support the key findings from simulations. As mentioned by the Reviewer, one of the most important design parameters for the confined water layer is the macrochannel diameter, which manipulates the strength of natural convection and enables the simultaneous salt rejection and thermal localization. In our original manuscript, we obtained the optimal macrochannel diameter through simulations and demonstrated good performance with this design. However, we did not experimentally show the impact of macrochannel diameter on the solar evaporation performance. In the revised manuscript, we provide additional experimental characterizations, which further confirm the optimized design parameters.

Specifically, we performed solar evaporation experiments of 20 wt% brine using confined water layer structures with 1 mm, 2.5 mm, and 5 mm macrochannel diameters, where we observed salt crystallization on the 1 mm diameter design and reduced thermal localization on the 5 mm diameter design. Both 1 mm and 5 mm diameter designs show lower solar-to-vapor conversion efficiency than the optimal design with 2.5 mm macrochannel diameter. In Page 10, Line 22 of the revised main text, we added the following discussion,

“We experimentally validated this regime by evaporating 20 wt% brine on three confined water layer structures with 1 mm, 2.5 mm, and 5 mm diameter macrochannels (see Supplementary Figure 6, 7 for details). Significant salt crystallization and increased heat loss through macrochannels were observed on the 1 mm and 5 mm diameter macrochannel designs, respectively, both of which reduce the solar-to-vapor conversion efficiency (see Supplementary Figure 7 for details). Guided by the modeling and experimental results, we chose $d = 2.5$ mm as the macrochannel diameter for our prototype.”

The corresponding experimental characterizations are shown in Supplementary Figure 6 and 7 in the Page 12 and 13 of the revised Supplementary Information,

Supplementary Figure 6. Top view of the (a) floating ring, insulation foams with (b) 1 mm, (c) 2.5 mm, and (d) 5 mm macrochannel diameters used for experimental characterizations. Unit is in mm.

Supplementary Figure 7. Solar evaporation of 20 wt% brine under one sun illumination. Three confined water layer structures with 1 mm, 2.5 mm, and 5 mm diameter macrochannels were tested (see Supplementary Figure 6 for detailed dimensions). (a) Temperature response of the confined water layer with different macrochannel diameters. The steady state temperature decreases with macrochannel diameters due to the increased heat loss through the floating insulation, which validates our theoretical prediction in Fig. 3a of the main text. (b) Corresponding mass change of the evaporation setup as a function of time. Confined water layer

structure with 2.5 mm diameter macrochannels shows the highest evaporation rate, which was chosen for our final design. The reduced evaporation rate of the 1 mm diameter macrochannel design is attributed to the salt crystallization and increased convective heat loss due to the elevated water layer temperature, whereas the reduced evaporation rate of the 5 mm diameter macrochannel design is induced by the increased conductive heat loss through macrochannels. Time-lapse images of the confined water layer structure with (c) 1 mm and (d) 5 mm diameter macrochannels during a 3-hour continuous solar evaporation. Salt crystallization was observed on the 1 mm diameter macrochannel design after the second hour, because natural convection enhanced salt rejection cannot be initiated with small macrochannel diameters. Scalebar: 2 cm.

4. Why did the authors choose such arrangement and number of the macrochannels in this manuscript?

Response: Thanks for this comment. We chose the current arrangement of macrochannels (*i.e.*, five macrochannels with 9 mm spacing) to ensure the uniform salt concentration in the confined water layer. For the same area coverage, 9 mm spacing between macrochannels is one of the most uniform arrangements, which can be useful to avoid locally concentrated brine. For the same average salt concentration, highly nonuniform salt concentration is undesirable, because salt crystallization always occurs at the position with the highest concentration.

To validate our macrochannel arrangement, we investigated the effect of macrochannel spacing on the nonuniformity of salt concentration $\Delta\phi_{\max}$, which is defined as the maximum value of salt concentration difference in the confined water layer. In general, macrochannel spacing weakly affects the salt transport whereas there is still an optimal spacing close to 9 mm for the highest salt concentration uniformity, which was chosen for our design. To make this point clear, in Page 11, Line 16 of the revised main text, we added the following discussion,

“Five macrochannels were fabricated where one of them was located at the center of the thermal insulation and the other four were 9 mm away from the center (see Supplementary Figure 2). We chose this configuration to further improve the uniform salt concentration in the confined water layer. With mechanistic modeling, we show that although macrochannel spacing plays a relatively insignificant role in salt transport, the 9 mm macrochannel spacing is desirable to enable the highest uniformity of salt concentration (see Supplementary Figure 9 for details).”

The corresponding detailed results are shown in Supplementary Figure 9 in Page 15 of the revised Supplementary Information,

Supplementary Figure 9. Effects of macrochannel spacing on the nonuniformity of salt concentration predicted by simulations. Solar evaporation of 20 wt% brine under one sun illumination was simulated. The macrochannel diameter is 2.5 mm and the confined water layer thickness is 5 mm. (a) Top view of three representative insulation foams with 4 mm, 9 mm, and 13 mm macrochannel spacing used for simulations. Unit is in mm. (b) Nonuniformity of salt concentration as a function of macrochannel spacing. In general, $\Delta\phi_{\max}$ weakly depends on the macrochannel spacing, but too large of a macrochannel spacing will lead to an increase of salt concentration nonuniformity. 9 mm spacing was chosen for our design to achieve the optimal macrochannel arrangement.

5. Does the wettability of the water pathway, i.e., the macrochannel in this system, influence the water transportation or confined water thickness, and further the performance of the solar evaporation system? In addition, the wettability of the polyurethane foam should be provided.

Response: Since both the passive water supply and confined water layer are realized by engineering the buoyancy of the neutrally floating thermal insulation, the wettability of the water pathway has negligible impact on the solar evaporation performance. Different from the wick-based solar evaporators, passive water supply of the confined water layer structure developed in this study does not rely on the capillary effect, for which reason, we call it as a “wick-free” design.

To better elucidate the wettability effect, we provide the following analysis. Specifically, the role of wettability in water transport can be quantified by the Young-Laplace equation,

$$\Delta p_{cap} = \frac{2\gamma}{R} \quad (R7)$$

where Δp_{cap} is the capillary pressure, γ is the surface tension of water (≈ 0.072 N/m), and R is the curvature of the liquid-air interface. For conventional wick-based solar evaporators, wicking occurs in micropores with high wettability, where a meniscus interface forms at the end of the microchannel (Fig. R11a). Since the curvature of the meniscus interface scales as the microchannel radius R_{wick} , typical $\sim 10 - 100$ μm radius microchannels will create $\sim 10^3 - 10^4$ Pa capillary pressure, which enables the passive water supply during solar evaporation. However, for our wick-free design, the liquid-air interface is on the top of the confined water layer (Fig. R11b) rather than the macrochannel or microchannel. The resulting curvature of the interface scales as the radius of the confined water layer R_{wick} ($\sim \text{cm}$). Even if the solid surface is superwetting, the corresponding capillary pressure is only ~ 1 Pa, which has negligible impact on the water supply. For an extreme condition with the confined water layer thickness approaching to zero, the liquid-air interface is at the end of the macrochannel. However, a superwetting 2.5 mm diameter macrochannel can only provide ≈ 100 Pa capillary pressure, which is insufficient for the passive water supply during solar evaporation. Therefore, it can be concluded that for the confined water layer structure, wettability has insignificant role in water transport. The passive water supply is mainly induced by the buoyancy effect.

To make this point clear, we added the following sentence in Page 6, Line 18 of the revised main text,

“Due to the neutral buoyancy and macrochannel connection, the self-floating thermal insulation moves synchronously with the water-air interface, maintaining a stable confined water layer throughout the entire evaporation process without the need for wicking.”

In fact, the polyurethane foam is not a highly wetting material, which further confirms that the wettability is not a significant parameter for the confined water layer structure design. The contact angle of a water droplet on the polyurethane foam is about 70° , which has been well-characterized by several studies [2-5]. As an example, for the convenience of the Reviewer, we copied the contact angle measurement results from Kuang *et al.*'s work to Fig. R12 of our Response [2].

Fig. R11. (a) Schematic of the conventional wick-based solar evaporator with meniscus liquid-air interface at the end of microchannels. (b) Schematic of the wick-free confined water layer structure developed in this work, where no meniscus interface forms.

Fig. R12. A representative contact angle measurement of polyurethane foam, which was copied from Fig. 2(a) of Kuang *et al.*'s work [2].

6. The thickness of the confined water layer, ~ 5 mm in this work, should be regulated and investigated as the water layer bottom is directly heated while evaporation occurs on its surface.

Response: We agree with the Reviewer that the impact of confined water layer thickness on both the thermal localization and salt transport should be investigated. In our work, the ~ 5 mm confined water layer thickness was obtained from a careful optimization. For the thermal localization, as expected by the Reviewer, too thick of a confined water layer will lead to (1) a large temperature gradient crossing the water layer and (2) an increase of heat loss through the sidewall. The uniformity of temperature distribution can be quantified by the Biot number Bi if there is heat conduction within the confined water layer only ($Bi = ht/k \approx 0.4 < 1$, where h is heat transfer coefficient, t is confined water layer thickness, and k is water thermal conductivity) [6]. For the 5 mm thickness, the Biot number of the confined water layer is approximately 0.4, indicating a reasonably good temperature uniformity. The natural convection in the confined water layer will further enhance the temperature uniformity. This combined effect has been considered in our mechanistic modeling, which couples the heat transfer with fluidic flow. In our original manuscript, we showed the highly uniform temperature distribution within the confined water layer using numerical simulation (Fig. R13a) and experimentally confirmed it with the IR image (Fig. R13b).

Fig. R13. (a) Simulated temperature distribution of the solar evaporation setup, which was copied from Fig. 4c of the main text. The numerical simulation considered the coupling of heat conduction and convection in water. The temperature in a 5 mm confined water layer (represented by the top rectangle in (a)) shows a highly uniform distribution. (b) Temperature distribution of the solar evaporation setup measured by an IR camera, which was copied from Fig. 4b of the main text. Temperature uniformity of the confined water layer was confirmed by this IR characterization.

In the revised manuscript, we further investigated the effect of confined water layer thickness on the total heat loss. Figure R14 shows the total heat loss increases with the confined water layer thickness due to the increase of the sidewall area. With the ≈ 5 mm thickness, the total heat loss through the confined water layer is reasonably low. Therefore, the confined water layer thickness used in our design ensures a reasonably good thermal localization.

Fig. R14. Heat loss as a function of the confined water layer thickness obtained from the mechanistic modeling.

On the other hand, in the revised manuscript, we also investigated the role of confined water layer thickness in salt transport. In general, although the confined water layer thickness has insignificant impact on the average salt concentration of the entire confined water layer, it strongly affects the uniformity of salt spatial distribution. Specifically, we show that a very thin confined water layer leads to highly nonuniform salt concentration with low concentration around the macrochannel and high concentration close to the external insulation ring (Fig. R15a). Since the salt transport resistance along the confined water layer $R_{s,\parallel}$ (in-plane direction) is inversely proportional to the confined water layer thickness t (i.e., $R_{s,\parallel} \sim 1/t$), reducing t increases $R_{s,\parallel}$ and hence leads to the highly nonuniform salt concentration (Fig. R15a). On the other hand, although increasing t is desirable for the in-plane salt transport, too thick confined water layer will increase the cross-plane transport resistance $R_{s,\perp}$ (i.e., $R_{s,\perp} \sim t$) and thus enlarge the nonuniformity of salt concentration crossing the confined water layer. The large thickness of confined water layer also increases heat loss due to the increase of sidewall area. The 5 mm thick confined water layer used in our design enables the highest uniformity of salt concentration (Fig. R15b) and avoids significant heat loss (Fig. R14).

Fig. R15. Salt concentration profiles in (a) 1 mm and (b) 5 mm thick confined water layers obtained from simulations. Macrochannel diameters for both cases are 2.5 mm and solar illumination is one sun. The salt concentration profiles were taken at ≈ 0.5 hours after the simulation started.

To elucidate the effect of confined water layer thickness, we added the following discussion in Page 11, Line 1 of the revised main text,

“Although the diameter of macrochannels is the most important design parameter to achieve the simultaneous thermal localization and salt rejection, careful optimization of the confined water layer thickness, macrochannel arrangement, and thermal insulation thickness was also performed to provide complete design guidelines for the confined water layer structure. In particular, the confined water layer thickness strongly affects the uniformity of salt concentration. Nonuniform salt concentration is undesirable for practical applications because salt crystallization will always occur at the position with the highest concentration. With mechanistic modeling of evaporating 20 wt% brine under one sun illumination, we showed that very thin confined water layer (< 3 mm) will lead to significant spatial nonuniformity of salt concentration (≈ 1 wt% salinity difference) due to insufficient in-plane salt transport. Meanwhile, too thick of a confined water layer (> 10 mm) will also increase the nonuniformity because of the reduced cross-plane salt transport (see Supplementary Figure 8a for details). In addition, the thick confined water layer will increase heat

loss due to the increase of sidewall area (see Supplementary Figure 8b for details). For these reasons, a 5 mm thick confined water layer was used in our design to ensure an optimal uniformity of salt concentration and insignificant heat loss.”

We also created a new Supplementary Figure (Supplementary Figure 8, Page 13) to support the discussion in the main text and provide detailed results,

Supplementary Figure 8. Effects of the confined water layer thickness on (a) salt transport and (b) heat loss predicted by simulations. Solar evaporation of 20 wt% brine under one sun illumination was simulated. The macrochannel diameter is 2.5 mm. (a) Nonuniformity of salt concentration $\Delta\phi_{\max}$ as a function of the confined water layer thickness at the transition point to the quasi-steady state (i.e., ≈ 0.5 hours after the simulation started). $\Delta\phi_{\max}$ is the maximum value of salt concentration difference in the confined water layer, which is defined as the difference between the highest salt concentration and the lowest salt concentration. Insets: top view of salt concentration profiles with 1 mm and 5 mm thick confined water layers. Nonuniform salt concentration is undesirable because salt crystallization will first occur at the position with the highest salt concentration. $\Delta\phi_{\max}$ first rapidly decreases and then slightly increases with the increase of the confined water layer thickness, leading to a minimum point approximately at 5 mm, which was selected for our design. The large nonuniformity of salt concentration occurs at a low confined water layer thickness (e.g., 1 mm), because the resistance of salt transport along the confined water layer (i.e., the lateral transport) is inversely proportional to its thickness, resulting in a low salt concentration near the macrochannel with a high salt concentration close to the external floating ring (see the inset of (a)). The slight increase in salt concentration nonuniformity at the large thickness (e.g., 10 mm) is attributed to the increased resistance of salt transport crossing the confined water layer (i.e., the vertical transport). (b) Heat loss through the sidewall of a floating ring as a function of the confined water thickness. Heat loss increases linearly due to the increase of sidewall area.

7. The locations of Fig. 3c and Fig. 3d are inconsistent with the description in the main text and the figure captions.

Response: We really appreciate the Review for pointing out the wrong figure order in our original manuscript. In the revised main text, we switched the order of the original Fig. 3c and d to make them consistent with the discussion in the main text and the corresponding captions. For the convenience of the Reviewer, we copied the revised Fig. 3 below.

Fig. 3 Fluidic flow engineered in the confined water layer. **a** Salinity ($\Delta\phi$) and temperature (ΔT) difference across the macrochannel as a function of the macrochannel diameter d . The sharp transition of $\Delta\phi$ and ΔT (marked by the blue and red dashed lines, respectively) indicates the convection dominated regime. The overlap of thermal localization (light red regime) and salt rejection (blue regime) represents of optimal design for the macrochannel diameter. **b** Confined water layer salinity ϕ as a function of time t after dripping 2.3 mL 20 wt% brine in an isothermal condition. ϕ was measured by a digital refractometer. Error bars represent the combination of instrumental uncertainty of digital refractometer (0.1 wt%) and the standard deviation of three-time measurements. Good agreement between the experiment and model prediction (red curve) is shown. **c** Time-lapse image of the transport of 20 wt% brine visualized by food dye. **d** Time-lapse

image of the transport of DI water visualized by food dye. For both visualization tests, the reservoir initially contained DI water only. The faster transport of 20 wt% brine (c) than DI water (d) confirmed the existence of the natural convection effect.

References for Response to Reviewer 3

1. Kuang, Y. *et al.* A High-Performance Self-Regenerating Solar Evaporator for Continuous Water Desalination. *Adv. Mater.* 31, 1900498 (2019).
2. Kuang, P. *et al.* Improved surface wettability of polyurethane films by ultraviolet ozone treatment. *Appl. Polym. Sci.* 118, 3024–3033 (2010).
3. Krol, P. & Krol, B. Surface free energy of polyurethane coatings with improved hydrophobicity. *Colloid Polym. Sci.* 290, 879–893 (2012).
4. Basturk, E., Madakbas, S., & Kahraman, M. V. Improved thermal stability and wettability behavior of thermoplastic polyurethane/barium metaborate composites. *Mater. Res.* 19, 434–439 (2016).
5. Hsu, S. H. *et al.* Evaluation and characterization of waterborne biodegradable polyurethane films for the prevention of tendon postoperative adhesion. *Int. J. Nanomedicine* 13, 5485 (2018).
6. Mills, A. F. *Heat Transfer*, Prentice Hall (1999).

REVIEWERS' COMMENTS

Reviewer #1 (Remarks to the Author):

well done

Reviewer #2 (Remarks to the Author):

I'm glad to see the authors have well addressed my previous comments with comprehensive and thorough revisions. This work should be accepted immediately for publication in Nature Communications.

Reviewer #3 (Remarks to the Author):

The authors have fully replied my comments and concerns, I think the revised manuscript is acceptable for publication.

Response Letter

Reviewer 1

well done

Response: We thank the reviewer for the positive assessment of the revised manuscript and for helping us improve the quality of our work.

Reviewer 2

I'm glad to see the authors have well addressed my previous comments with comprehensive and thorough revisions. This work should be accepted immediately for publication in Nature Communications.

Response: We really appreciate the Reviewer for the valuable comments and careful review of our revised manuscript. We are glad that the Reviewer believes all the previous comments have been well addressed.

Reviewer 3

The authors have fully replied my comments and concerns, I think the revised manuscript is acceptable for publication.

Response: We thank the Reviewer for providing useful suggestions which helped us improved the manuscript. We are glad that the Reviewer believes our work now can be accepted.